# Melatonin Prevents Depression but Not Anxiety-like Behavior Produced by the Chemotherapeutic Agent Temozolomide: Implication of Doublecortin Cells and Hilar Oligodendrocytes

**DOI:** 10.3390/ijms241713376

**Published:** 2023-08-29

**Authors:** Edith Araceli Cabrera-Muñoz, Gerardo Bernabé Ramírez-Rodríguez, Lizeth Díaz-Yañez, Verónica Reyes-Galindo, David Meneses-San Juan, Nelly Maritza Vega-Rivera

**Affiliations:** 1Laboratorio de Neurogénesis, Subdirección de Investigaciones Clínicas, Instituto Nacional de Psiquiatría “Ramón de la Fuente Muñiz”, Calzada Mexico-Xochimilco 101, Ciudad de México 14370, Mexicomeneses.sj.d@outlook.com (D.M.-S.J.); 2Instituto de Ecología, Universidad Nacional Autónoma de México, Circuito Exterior sin Número, Ciudad Universitaria, Ciudad de México 04510, Mexico; 3Laboratorio de Neurpsicofarmacología, Dirección de Neurociencias, Instituto Nacional de Psiquiatría “Ramón de la Fuente Muñiz”, Calzada Mexico-Xochimilco 101, Ciudad de México 14370, Mexico; nmvega@imp.edu.mx

**Keywords:** melatonin, temozolomide, oligodendrocytes, hippocampus, depression, anxiety, adult neurogenesis

## Abstract

Melatonin is a hormone synthesized by the pineal gland with neuroprotective and neurodevelopmental effects. Also, melatonin acts as an antidepressant by modulating the generation of new neurons in the dentate gyrus of the hippocampus. The positive effects of melatonin on behavior and neural development may suggest it is used for reverting stress but also for the alterations produced by chemotherapeutic drugs influencing behavior and brain plasticity. In this sense, temozolomide, an alkylating/anti-proliferating agent used in treating brain cancer, is associated with decreased cognitive functions and depression. We hypothesized that melatonin might prevent the effects of temozolomide on depression- and anxiety-like behavior by modulating some aspects of the neurogenic process in adult Balb/C mice. Mice were treated with temozolomide (25 mg/kg) for three days of two weeks, followed by melatonin (8 mg/kg) for fourteen days. Temozolomide produced short- and long-term decrements in cell proliferation (Ki67-positive cells: 54.89% and 53.38%, respectively) and intermediate stages of the neurogenic process (doublecortin-positive cells: 68.23% and 50.08%, respectively). However, melatonin prevented the long-term effects of temozolomide with the increased number of doublecortin-positive cells (47.21%) and the immunoreactivity of 2′ 3′-Cyclic-nucleotide-3 phosphodiesterase (CNPase: 82.66%), an enzyme expressed by mature oligodendrocytes, in the hilar portion of the dentate gyrus. The effects of melatonin in the temozolomide group occurred with decreased immobility in the forced swim test (45.55%) but not anxiety-like behavior. Thus, our results suggest that melatonin prevents the harmful effects of temozolomide by modulating doublecortin cells, hilar oligodendrocytes, and depression-like behavior tested in the forced swim test. Our study could point out melatonin’s beneficial effects for counteracting temozolomide’s side effects.

## 1. Introduction

Diagnosis and treatment are determining factors for the treatment of cancer [1,2,3,4,5,6,7]. In the case of brain cancer, some medications are better able to pass through the blood–brain barrier, such as temozolomide (TMZ), an alkylating agent used with daily low-dosages initially in combination with radiotherapy, followed by monthly doses of TMZ alone for long periods (6–12 months) [1,2,3,4,5,6,7]. TMZ is indicated for glioblastoma multiforme, the most frequent primary brain cancer, with a higher incidence in men than women. However, secondary glioblastoma tumors, which progress from low-grade diffuse astrocytoma or anaplastic astrocytoma, are seen more frequently in women [1,2,3,4,5]. TMZ shows promising efficacy in treating brain cancer, but patients treated with TMZ report depressive episodes and progressive decreased cognitive abilities [1,2,3,4,5,6,7]. Recently, in C57Bl/6J mice, the cyclic treatment administered in the clinic with daily injections of TMZ on three consecutive days every week was mimicked, causing the number of new neurons in the dentate gyrus (DG) of the hippocampus to diminish, and this decrease was associated with depressive-like behavior [8].

Melatonin (MEL), the primary hormone produced by the pineal gland, has been associated with antidepressant, neuroprotective, and neurogenic effects [9,10,11,12,13,14,15,16]. Several studies reveal the role of MEL in depression-related symptoms, such as diminished cognitive ability and difficulties in falling asleep [17,18]. Interestingly, low levels of this hormone in patients with depression have been observed [13]. The patients who received the conjunct treatment of antidepressants and MEL showed better results than those treated only with antidepressants [19]; those results are replicated in animal models with several strains of mice [14,16,20]. Although the mechanisms implicated in these melatonin effects are unknown, they could implicate the regulation of brain plasticity involving neurons and glial cells, as reported recently [15,21].

In the present study, as a starting point to evaluate the benefits of MEL on the deleterious effects of TMZ on some events of the neurogenic process and behavior, we used female Balb/C mice. Thus, we hypothesized that MEL might prevent the effects of TMZ on depression- and anxiety-like behavior through the modulation of cell proliferation and doublecortin cells in female adult Balb/C mice. Previous studies on female and male rodents have indicated that MEL regulates the cell types located at the neurogenic niche. For instance, MEL increases cell proliferation, cell survival, neuronal differentiation, and the axons of new and old neurons but also modulates microglia [14,15,16]. In addition, other studies have proven the capability of MEL to regulate oligodendrocytes, cells relevant for the myelination of axons, as well as the modulation of neuronal function, and to provide neuronal trophic support through the production of soluble factors in the neurogenic niche of the DG in the hippocampus [12,22,23]. Here, we analyzed the short- and long-term effects of TMZ on cell proliferation and intermediate stages of the neurogenic process in the DG of the hippocampus. Also, we examined the effects of MEL after the TMZ treatment at the level of cell proliferation (Ki67), doublecortin (DCX), mossy fiber volume (calbindin), and CNPase immunoreactivity in the DG and hilar zone in Balb/C mice.

## 2. Results

### 2.1. Temozolomide-Induced Anxiety-like Behavior, Which Melatonin Does Not Prevent

First, we explored the long-term effects of TMZ on behavior (Figure 1A). We first exposed mice to the rotarod to discard motor coordination alterations produced by treatments. We did not find differences among the groups (one-way ANOVA: F3,29 = 0.10, *p* = 0.95), suggesting that temozolomide, melatonin, or the vehicle were not affecting locomotor activity. Then, the open field test (Figure 1B,C) showed that mice treated with TMZ showed more time in the periphery (Figure 1B: F3,26 = 12.95, *p* < 0001) and less time in the center (Figure 1C: F3,26 = 40.25, *p* < 0001) than the control (*p* = 0.017, *p* < 0.001, respectively) and MEL (*p* < 0.001 for both parameters, respectively) groups. Mice that received MEL after TMZ did not show modifications in the time in the periphery (*p* = 1) or the center (*p* = 1) compared with the TMZ group.

Also, mice were tested in the elevated plus maze (EPM). The EPM test (Figure 1D–G. Closed arms (events): F3,27 = 0.48, *p* = 0.69; closed arms (time): F3,27 = 5.45, *p* = 0.005; opened arms (events): F3,27 = 7.83, *p* < 0.001; and opened arms (time): H = 17.29, d.f. = 3, *p* < 0.001) revealed that TMZ compared with the control group did not modify the number of events (Figure 1D, *p* = 0.49) or the time in the closed arms (Figure 1E, *p* = 1). Similar effects of TMZ were seen in the number of events in the open arms (*p* = 1, Figure 1F), but the time spent in the open arms was lower than in the control group (q = 4.78, *p* < 0.05, Figure 1G), whereas MEL-treated mice showed increased time spent in the opened arms (q = 4.9, *p* < 0.05) and decreased events in the opened arms (Figure 1F, *p* = 0.006) compared with the control group. But, the time spent in the open arms was higher in the MEL-treated group compared with the TMZ group (q = 6.12, *p* < 0.05, Figure 1G) or in TMZ followed by MEL (q = 5.23, *p* < 0.05). Also, the number of events in the open arms was higher in the TMZ than in the MEL group (*p* = 0.002, Figure 1F), and it showed a trend toward a significant difference (*p* = 0.052) compared with mice that received TMZ followed by MEL. Mice treated with TMZ did not show differences in the number of events in the closed arms compared with the other groups (Figure 1D). Also, the analysis of the time spent in the closed arms (Figure 1E) showed similar results among groups, although MEL significantly increased the time spent in the closed arms than TMZ and control (*p* = 0.042, *p* = 0.005, respectively). Interestingly, TMZ and TMZ followed by MEL did not show differences (*p* = 1). Thus, we calculated the anxiety index (AI) (Figure 1H: F3,27 = 6.48, *p* = 0.002). The AI confirmed that melatonin could not prevent the anxiety-like behavior produced by TMZ (*p* = 0.003). These results suggest that MEL does not prevent anxiety-like behavior caused by TMZ.

### 2.2. Temozolomide-Induced Depression-like Effects, Which Melatonin Prevents

Also, we evaluated the effects of TMZ on despair-like behavior in the forced swim test (FST) (Figure 1H). TMZ increased the immobility time in the FST (Figure 1H: F3,26 = 11.54, *p* < 0.001) compared with MEL (*p* < 0.001), with MEL after TMZ treatment (*p* < 0.001), or with the control group (*p* = 0.029), respectively. These results suggest that MEL prevented depression-like behavior but not anxiety-like behavior in the open field test caused by TMZ.

### 2.3. Temozolomide Induced Short- and Long-Term Effects on Cell Proliferation and Intermediate Stages of Hippocampal Neurogenesis

A recent study reported that TMZ induces depression-like behavior and alterations in the hippocampal neurogenic process [8]. Thus, we initially analyzed the short- and long-term effects of TMZ on Ki67 and DCX cells in the subgranular- and granular-cell layers in the DG of the hippocampus, respectively, (Figure 2) in rodents that did not go to the behavioral tests. Cellular quantifications were performed the day after the last administration of TMZ or 14 days after the TMZ treatment finished (Figure 2A). Two-way ANOVA for Ki67 cells showed the main effect of treatment (F1,11 = 41.265, *p* <0.001) but not at the time of evaluation (F1,11 = 0.569, *p* = 0.47). Thus, proliferative cells in mice treated with TMZ decreased at both assessment time points without showing the interaction of both factors (F1,11 = 1.84, *p* = 0.21: Figure 2B,C). Similar effects were seen after the quantification of DCX cells. TMZ produced a short- and long-term decreased number of DCX cells (main effect of treatment (F1,11 = 38.72, *p* < 0.001) but not at the time of evaluation (F1,11 = 1.8, *p* = 0.21). Again, DCX cells in mice treated with TMZ decreased at both assessment time points without showing an interaction of both factors (F1,11 = 4.16, *p* = 0.076: Figure 2D,E).

### 2.4. Melatonin Prevents Temozolomide’s Long-Term Effects on Intermediate Hippocampal Neurogenesis Stages

Once we observed the short- and long-term effects of TMZ on Ki67 and DCX cells (Figure 2), we analyzed whether MEL could prevent the long-term effects of TMZ on proliferative (Ki67) and DCX cells in female Balb/C mice that underwent behavioral tests (Figure 3). Cellular quantification of Ki67 cells showed an increasing number in MEL-treated mice (31.90 ± 0.63, *p* < 0.001 versus control group) and a reduction in the TMZ group (6.87 ± 0.68, *p* < 0.001 versus control group), compared to control mice (21.39 ± 0.69). Mice treated with MEL after TMZ showed a similar number of Ki67 cells to mice treated with TMZ (TMZ + MEL: 8.93 ± 0.70, *p* = 0.07) (Figure 3A,B. Ki67: F3,18 = 249.61, *p* < 0.001).

Quantification of DCX cells indicated that MEL increases (MEL: 41.93 ± 2.28; *p* = 0.002 vs. control group), but TMZ reduces (TMZ: 19.21 ± 1.2; *p* < 0.001 vs. control group) the number of DCX cells compared with the control group (31.21 ± 2.03) (Figure 3C,D). However, mice treated with MEL after TMZ showed an increased number of DCX cells compared with mice treated with TMZ (28.28 ± 082; *p* = 0.007) (Figure 3C,D. DCX: F3,17 = 26.77; *p* < 0.001). These results suggest that MEL prevented the effects of TMZ on intermediate stages of adult neurogenesis.

### 2.5. Melatonin Increased the Volume of the Infrapyramidal Mossy Fibers Tract in the Hilus of the Dentate Gyrus

After we observed that MEL seems to prevent the effects of TMZ on the decreased number of DCX cells (Figure 3), that axons of newborn neurons contribute to mossy fibers, which are implicated in several types of behavior [24], and that the indole can increase the number of DCX cells and the volume of mossy fibers [14], we quantified the volume of the supra- and infrapyramidal mossy fibers through the identification of calbindin (Figure 4), as was previously reported [14,24,25].

The immunoreactivity of calbindin in the infrapyramidal mossy fibers showed that MEL increased the volume of the infrapyramidal mossy fiber tract (498,150 ± 5694 cubic micrometers, *p* < 0.005 vs. control group). Still, mice of the TMZ (302,043 ± 21,893 cubic micrometers, *p* = ns) and TMZ + MEL (242,514 ± 13,673 cubic micrometers, *p* < 0.005) groups showed decreased volume compared to the control group (361,739 ± 13,053 cubic micrometers). However, the decrement observed in the TMZ + MEL group was statistically significant (Figure 4B, *p* < 0.03 vs. the control group (F3,11 = 55.13, *p* < 0.0001)). However, the treatments did not affect the volume of the suprapyramidal mossy fibers (Figure 4C (F3,11 = 0.83, *p* = 0.50)).

These results confirmed that MEL increases the volume of the infrapyramidal mossy fiber tract, but the indole could not prevent the reduction induced by TMZ.

### 2.6. Temozolomide Decreased CNPase Immunoreactivity in the Hilus but Not in the Granular Cell or the Dentate Gyrus’s Molecular Layer

Because MEL increased the number of DCX cells and the volume of the infrapyramidal mossy fibers, we quantified the immunoreactivity of CNPase (Figure 5), a marker for mature oligodendrocytes, in the granular cell layer (GCL: Figure 5B), molecular layer (ML: Figure 5C), and the hilus of the DG (Figure 5D). From the three regions analyzed, we found that only MEL by itself did not increase the immunoreactivity of CNPase compared to the control group (*p* = 0.46) (F3,11 = 89.7, *p* < 0.001). However, TMZ decreased CNPase immunoreactivity in the hilus (829,297 ± 78,108 O.D.) compared to the control group (2,089,235 ± 24,381; *p* < 0.001; Figure 5D). Interestingly, MEL prevented the decreased immunoreactivity of CNPase caused by TMZ (1,514,817 ± 53,783 O.D.; *p* < 0.001; Figure 5D). However, in the GCL or the ML there were no differences among the groups (F3,11 = 1.94, *p* = 0.16 and F3,11 = 2.22, *p* = 0.14, respectively).

### 2.7. Principal Component Analysis

Finally, we performed the principal component analysis (PCA, Figure 6). We assembled a comprehensive set of five behavioral and four physiological parameters (Methods). The PCA revealed two principal eigenvalues greater than 1 (Table 1). The PCA gave two main components explaining the 81.9% variation (PC1 62.9%, PC2 19%) (Table 2: Figure 6). The PCA confirms the effect of treatment on behavioral and physiological values (Figure 6). Ki67 cells, infrapyramidal mossy fibers (CB IFPMF), time in the center of the open field (OF center time), time in the periphery of the open field (OF periphery time), DCX cells, and the immunoreactivity of CNPase in the hilus loaded strongly and concordantly on PC1 (62.9%, Table 2), indicating that PC1 primarily represents variation in anxiety behavior and proliferative and immature cells but also CNPase immunoreactivity in the hilus. PC2 (19%, Table 2) represents FST immobility time and EPM open arms time; this may indicate a change in behavior with the TMZ–MEL treatment. The Kaiser–Meyer–Olkin (KMO) revealed the appropriate use of the factorial analysis (0.51). Also, Bartlett’s test of homogeneity of variances was significant (BKs = 134.1, d.f. = 8, *p* < 2.2 × 10^−16^), suggesting a significant and relevant correlation between the variables and the factor model.

## 3. Discussion

In this study, we confirmed that TMZ, an alkylating agent for treating brain cancer, especially glioblastoma [2,3,26], causes depressant- and anxiety-like behavior, but it also decreased the number of Ki67 and DCX cells in rodents [8]. In addition, we observed that TMZ decreased the immunoreactivity for CNPase in the hilus. Those effects of TMZ on depressive-like behavior, Ki67 cells, DCX cells, and the immunoreactivity of CNPase were prevented with the administration of MEL in female Balb/C mice.

### 3.1. Temozolomide Induces Behavior Related to Depression and Anxiety, but Melatonin Only Prevented Depression-like Behavior

TMZ produced depressant-like behavior in male C57Bl6 mice [8]. Here, we confirmed the depressive-like behavior induced by TMZ but in female Balb/C mice exposed to the forced swim test. This paradigm lets us explore despair or the inability to attempt escape, which is suggested to be an indicator of depressive-like behavior [27]. Also, we observed that TMZ increased the time spent in the periphery of the open field and that there was a slight decrease in the time invested in open arms in the EPM. These results are similar to those found in male C57Bl6 mice [8]. Interestingly, previous studies suggested the benefits of MEL to prevent or reverse depressive-like behavior in C57Bl6 mice exposed to the FST and Balb/C mice chronically stressed, respectively, [15,27,28]. Here, MEL decreased the total immobility time, suggesting that those animals that faced the forced swim test better coped with a stressor that shows behaviors like depression. Also, MEL decreased the anxious-like behavior, increasing the time spent in the center during the open field and the open arms of the elevated plus maze [28,29,30,31,32,33,34,35]. Thus, our results suggest that MEL has antidepressant- and anxiolytic-like effects, as was previously reported [20,28,29,34,35]. However, the effects of TMZ on the increased immobility in the FST were prevented by MEL. However, the indole failed to prevent the anxiogenic effect of TMZ in female Balb/C mice. Here, the antidepressant-like effects of MEL were evaluated using the FST. The use of this approximation has been recently debated, pointing to the necessity of validating the antidepressant effects seen in the FST with other approaches [27,36]. However, the antidepressant-like effect of MEL and the depressive induction behavior with TMZ has been confirmed in the chronic mild stress model [35] and the novelty-suppressed feeding test, respectively, [8]. Thus, we consider that the effect of MEL and TMZ on depressive-like behavior is supported by previous studies, i.e., [8,15].

The effects of MEL to prevent alterations produced by TMZ on depression-like behavior may implicate the participation of its membrane receptors known as melatonin receptor 1 (MT1) and melatonin receptor 2 (MT2). These receptors are expressed in the DG of rats [37] but also in the DG of C57Bl6 mice [28]. The relevance of both receptors for MEL effects on depression- and anxiety-like behaviors were proved in C3H/Hen MT1 or MT2 knockout mice [38,39,40]. MT1 knockout mice exhibited increased immobility related to the despair behavior, suggesting this receptor’s relevance for the effects of MEL on depression-like behavior [38]. Here, we did not find results of MEL on anxiety behavior caused by TMZ, but recent studies support the relevance of the MT2 receptor for the anxiolytic effects of MEL [39,40]. Thus, our study suggests that in TMZ-treated mice, MEL can prevent depressive-associated behavior, probably through the activation of MT1 receptors.

### 3.2. Melatonin Prevents the Decrement of Ki67 and Doublecortin Cells in the Granular Cell Layer and the CNPase Immunoreactivity in the Dentate Gyrus’s Hilus Caused by Temozolomide

Depression- and anxiety-like behaviors are linked to a low level of neurogenesis in the hippocampus [41,42,43,44]. In this regard, TMZ decreased Ki67 and DCX cells in male and female C57Bl6 mice [8,45]. In those studies, TMZ was given for four and six weeks following an intermittent administration protocol [8,45]. Still, the cell proliferation and DCX cells were measured directly from the TMZ administration or after a six-week recovery period [8]. Here, we administered TMZ for two weeks. The effect of TMZ on the decreased number of Ki67 and DCX cells lasted for 19 days, although the more significant impact was seen on the number of DCX cells, while MEL had the contrary effect, increasing them. In this sense, a previous study of our group reported that MEL favored the generation of DCX-immature cells together with the increased complexity of the dendrite tree and antidepressant-like effect [46]. Also, we previously reported that endogenous MEL could influence cell proliferation (Ki67) and proliferative DCX cells, results that were prevented with the chronic administration of luzindole, an antagonist of MT1 and MT2 receptors, without affecting the dendrite maturation of DCX cells in C57Bl6 mice [10]. Thus, the results of the present study suggest that MEL mainly prevents the effects of TMZ on the decreased number of DCX cells. However, we cannot discard other processes related to neuroplasticity. For instance, melatonin favors the maturation of dendritic spines, which was not addressed in the present study, and also axonal maturation and the participation of oligodendrocytes [47,48].

Moreover, it is known that the maturation of new neurons involves the axonal projection through the mossy fibers essential to establishing functional connectivity to CA3. It is suggested that this projection goes through the infrapyramidal blade of GD, then connects to CA3 with the pyramidal neurons in the hippocampus [24,49,50,51,52,53]. Also, a previous study showed that the infrapyramidal mossy fibers are influenced by increased neurogenesis, and newborn neurons’ axons contribute to the infrapyramidal projection [24]. Interestingly, mossy fibers coincide with better performance on several behavioral tests [52,53]. In the context of the pathophysiology and treatment of neuropsychiatric disorders [54], mossy fibers show relevance for the connection to CA3. Although we observed that TMZ decreased the volume of mossy fibers projection, it was not prevented with the administration of MEL.

Nevertheless, we confirmed that MEL modifies the axonal structure by increasing the volume of mossy fibers in the infrapyramidal blade in rodents only treated with indole [14]. Interestingly, the results of this study suggest that the antidepressant-like effect of MEL is not influenced by increasing the volume of mossy fiber projection. However, the increased volume of mossy fibers produced by MEL may be related to the participation of the MT2 receptor that is essential for the axonogenesis of hippocampal neurons in vitro and in vivo [55]. The effects of MEL mediated through the activation of the MT2 receptor for axonogenesis involves the v-Akt Murine Thymoma Viral Oncogene/Protein Kinase-B), a serine/threonine protein kinase, also known as Akt, engaged in proliferation and survival [55]. Here, the increased volume of the infrapyramidal mossy fibers could occur through the activation of the MT2 receptor. However, this assumption needs to be probed in an additional study using specific inhibitors or knockout mice, as was conducted to test the antidepressant- or anxiolytic- effect [37,38,39].

Interestingly, mossy fibers pass through the hilus to establish functional connectivity to CA3. In the hilus, oligodendrocytes and their precursors are critical facilitators of neurophysiology [56]. They provide trophic and metabolic support to neurons [57,58]. Recent studies have pointed to the relevance of oligodendrocytes in depression [59]. Some studies reported that MEL increased the oligodendrocyte differentiation from neural stem cells (NSC) [48,60]. Embryonic mouse-derived neural stem cell culture in the presence of MEL for five days showed an increased percentage of myelin basic protein-positive cells (MBP-cells) [48,60]. Here, the effects of TMZ and MEL in TMZ-treated mice only appeared in the hilus but not in the molecular or granular layers of the DG. Our study identified oligodendrocytes through the immunoreactivity of CNPase, a protein expressed during the transition from progenitor to differentiated oligodendrocytes. This protein has two different isoforms (short and long isoforms), which are differentially expressed during oligodendrocytes’ formation [61,62]. Here, TMZ decreased the immunoreactivity of CNPase, but the decrement was prevented with the administration of MEL. However, the indole alone did not modify the CNPase immunoreactivity. This result is in line with a study performed on rats with damage in the white matter. There, MEL promoted the maturation of oligodendrocytes after injury but not in healthy subjects [12]. Also, MEL promotes myelination in a postnatal stroke model [22]. Thus, our results suggest that the effects of MEL to prevent the alterations produced by TMZ also involved modifications in the oligodendrocytes.

According to the results obtained, we propose that MEL could exert its effects on depressive behaviors caused by TMZ through plastic changes in the brain, beyond the intermediate stages of the hippocampal neurogenic process, as an isolated process. The effects of MEL on the recovery of oligodendrocytes could positively contribute to providing trophic factors necessary for neuronal development or the establishment of synapses with CA3 [22]. All this together could regulate the antidepressant effects of MEL (Figure 7).

Interestingly, the results obtained from our PCA analysis suggest that the most significant contributions to the impact of TMZ and MEL on behavior were those of DCX cells, followed by the immunoreactivity of CNPase in the hilus. Although the participation of microglia needs to be considered, a previous study reported that TMZ is not affecting microglia [45].

### 3.3. Conclusions and Limitations

We considered that our study supports the relevance of MEL to prevent alterations caused by TMZ on behavior and plasticity in female Balb/C mice, pointing to the direction of the possible use of MEL to relieve the side effects of TMZ.

Nevertheless, a limitation of our study is that we did not use a murine cancer model, and our results need to be replicated in a preclinical study using cancer brain animal models, including female and male mice. Also, we considered that it is necessary to explore the mechanism underlying the antidepressant-like effect of MEL involving more than one single-cell type in neurogenesis and the participation of membrane receptors, and the interaction with cytoskeleton-associated proteins is necessary to understand the relevance of MEL in physiological and non-physiological conditions.

## 4. Materials and Methods

### 4.1. Animals

Balb/C female mice were ten weeks old, with 20 ± 2 g used in this study. They were held in standard laboratory plexiglass cages (L × W × H: 48 × 39 × 20 cm) with 12 h light/12 h dark cycles (Zeitgeber time 0: ZT0) at 0700 h and to the timing of lights off (Zeitgeber time 12: ZT12) at 1900 h, respectively, at a temperature of 23 ± 1° C in the animal facilities of the National Institute of Psychiatry Ramón de la Fuente Muñiz. During the time of housing, mice had a piece of paper as nesting material, which was supplied twice a week. Mice had access to food and water ad libitum, and handling procedures complied with the Mexican Official Standard for animal care (NOM-062-ZOO-1999) and were approved by the local Institutional Ethics Committee of the National Institute of Psychiatry “Ramón de la Fuente Muñiz.” All efforts were made to minimize animal suffering and to reduce the number of animals. In this study, we used females instead of male mice due to the territorial effect present in males. In addition, female mice show less aggressive behavior than males after being randomly assigned to the experimental groups [63]. This strain of mice was also chosen due to its sensitivity to stress and its response to chronic antidepressants.

### 4.2. Melatonin, Temozolomide, and Experimental Design

Melatonin (Sigma, Naucalpan, Estado de México, México) was freshly prepared daily and dissolved in a minimum volume of pure ethanol plus saline solution (0.9% NaCl), with a final working concentration of 8 mg/kg body weight. The final volume of ethanol in the melatonin vehicle was less than 1% in a ratio of 1:100 (ethanol:saline solution) [20]. TMZ (Sigma, Naucalpan, Estado de México, México) was prepared in saline solution (PISA, Guadalajara, Jalisco, México) with a final 25 mg/kg concentration. The TMZ treatment followed a discontinuous cyclic administration as in preclinical studies, showing depressive-like behavior and decreased hippocampal neurogenesis [8,45].

#### 4.2.1. Experiment 1 (Figure 1, Figure 3, Figure 4, Figure 5 and Figure 6)

Mice were randomly distributed into four groups: (1) CTR (saline solution + VEH), (2) TMZ (TMZ + VEH), (3) MEL (saline solution + MEL), and (4) TMZ plus MEL (TMZ + MEL). Saline solution or TMZ was injected intraperitoneally for three consecutive days for two weeks (D1 to D3, D8 to D10), followed by VEH or MEL intraperitoneally for 14 days (D11 to D24). Forty-eight hours after the last administration, rodents underwent behavioral tests.

#### 4.2.2. Experiment 2 (Figure 2)

To test the short- and long-term effects of TMZ on cell proliferation and intermediate stages of the hippocampal neurogenic process, Balb/C mice were distributed into four groups. The first two groups were control short-term (CTR short-term) or TMZ short-term. Mice were treated with saline or TMZ, injected intraperitoneally for three consecutive days for two weeks (D1 to D3 and D8 to D10). Thus, 24 h later the mice were euthanized at D11. The long-term groups were control long-term (CTR long-term) or TMZ long-term. Therefore, mice were treated with saline, injected intraperitoneally for three consecutive days for two weeks (D1 to D3 and D8 to D10) and euthanized fourteen days after the last administration of TMZ (D25); animals were euthanized, and the brain was collected and maintained for subsequent histological evaluations.

### 4.3. Behavior

After treatments, from D26 to D29, mice were exposed to a rotarod (D26) and a battery of behavioral tests (open field, elevated plus maze, and Porsolt’s forced swimming test, respectively). Before each behavior assay began, animals were habituated to the testing room for 30 min. Behavioral testing was performed at the end of the dark phase of the light/dark cycle (ZT23). Animals were naïve to the behavioral tests.

#### 4.3.1. Rotarod and Open Field Test

A rotarod test was performed in a touchscreen rotarod apparatus (Panlab, Cornelia, Barcelona, Spain) to evaluate the TMZ effects on the motor coordination of mice [30]. Mice were gently held by the tail on the rotarod, facing the rotation sense to walk easily. Speed started at 4 rpm constantly for 10 s, and then an acceleration rate of 20 rpm/min was applied. Two sessions were measured with 10 min between them, and the rate was reported.

The exploration and anxiety were measured by an open field test [64]. Briefly, large boxes (34 × 44 × 20 cm) were divided into 16 squares to delimit the center (4 squares) and the periphery (12 squares). For each testing session, a mouse was gently placed individually in the arena’s corner and allowed to explore the environment for 5 min freely.

#### 4.3.2. Elevated Plus Maze

To evaluate the anxiety parameter, the elevated plus maze test was applied. Mice were tested according to previous protocols [32,33]. The maze apparatus was acrylic and consisted of two opposing closed arms, which were enclosed by 15 cm high grey acrylic walls, and two opposing open runways; the maze was elevated 40 cm from the ground. Mice were placed on the central platform, facing the open arms. Exploration along the maze was recorded for 5 min. Behaviors assessed in the EPM are time spent (duration) in open and closed arms, the number of open and closed arms entries (events), and the total exploration (entries into all arms). The total exploration is calculated as the number of entries into any arm of the maze to distinguish between poor exploratory behavior, exploration limited to closed arms (avoidance), and free exploration. Some of the parameters mentioned above are used to calculate an anxiety index, which is calculated as follows: anxiety index = 1 − [(time spent in the open arms/total time on the maze) + (number of entries to the open arms/total exploration on the maze)/2]; the anxiety index values range from 0 to 1 where an increase in the index expresses increased anxiety-like behavior. The behavior of the animals was video recorded and manually analyzed blind to the experiment.

#### 4.3.3. Porsolt’s Forced Swimming Test

A forced swimming test was applied per the previous protocol [12,28]. Mice were gently placed in a cylinder (15 cm in diameter) filled to a depth of approximately 15 cm with water maintained at room temperature for 6 min. Afterward, each mouse was gently dried with a paper towel, placed in a heated cage with standard bedding covered by an absorbent paper towel for 10 min, and returned to their home cages for an additional 50 min before being euthanized. Each session was videotaped for 6 min to their posterior analysis; the first minute was considered a habituation period to the test, and the last 5 min were analyzed. The behavior parameter investigated was immobility detection (assuming the minimal movements exerted by the animal to keep its head above water and floating). Thus, we studied the latency to the first immobile episode, the number of bouts of immobility, and the time spent immobile. The period considered for immobility was 3000 milliseconds. The behavior of the animals was video recorded and analyzed blind to the experimental group with the ANY-maze behavioral tracking software (Stoelting Co., Wood Dale, IL, USA).

### 4.4. Brain Tissue Processing

The animals were euthanized by decapitation two hours after the behavioral test, and the brain was collected for the histological procedure or protein determination. Brains destined for histology were immersed in 4% p-formaldehyde and kept for seven days. Then, brains were cryoprotected with sucrose solution (30%) and stored at 4 C [14,26].

### 4.5. Histology

Brains were cut into 40 μm coronal sections on a sliding microtome (Leica, Buffalo Grove, IL, USA). The brain slice sections were stored at 4 °C in a cryoprotective solution containing 25% ethylene glycol and 25% glycerol in 0.05 M phosphate buffer [14,15,16]. Brain slices were washed with 1× Tris-Buffered Saline (TBS) for 15 min and passed with 0.6% H_2_O_2_ (Sigma, Naucalpan, Estado de México, México) solution for 30 min. Next, brain slices were treated with citrate buffer solution (pH 6.0) at 90 °C for 30 min and then at 4 °C for 20 min. After a 15 min wash in 1× TBS, the tissue was placed in a blocking solution (1% Triton X-100 (Sigma, Naucalpan, Estado de México, México), 3% goat/donkey serum (Jackson Immunoresearch, West Grove, PA, USA), and 96% 1× TBS for 30 min. Different markers were evaluated to analyze some events of the neurogenic process, such as cell proliferation (Ki67) and intermediate stages of neuronal development (DCX). To visualize axons, we identified Calbindin projections (Calbindin, Cal). For oligodendrocytes, we identified CNPase. Brain coronal sections were incubated with primary antibodies to detect Ki67, DCX, Cal, or CNPase, and the peroxidase DAB method was performed [14,15,16]. Positive cells for Ki67 and DCX were quantified in every sixth section from all animals and visualized using a 40× objective throughout the rostrocaudal axis. Counting was performed using the modified optical dissector method under bright-field light microscopy (Leica, Buffalo Grove, IL, USA). The cells appearing in the uppermost focal plane were excluded to avoid over-sampling. Positive cells and immunoreactivity were quantified in the dentate gyrus (DG) (Bregma −1.34 to −3.20). The antibodies used were rabbit anti-Ki67 (1:1000; ab15580, Abcam, San Francisco, CA, USA), goat anti-DCX (1:500; sc-8066, Santa Cruz Biotechnology, Dallas, TX, USA), rabbit anti-Calbindin (1:2000; CB-38a, Swant, Burgdorf, Switzerland), and mouse anti-CNPase (1:500; ab6319, Abcam, San Francisco, CA, USA). For the mossy fiber projection (the suprapyramidal- and infrapyramidal-mossy fiber projections) volume, it was estimated as we reported previously [24] using the Cavalieri principle in coronal 40 μm sections, 240 μm apart, covering the complete DG in its rostrocaudal extension in CTR, TMZ, MEL, and TMZ + MEL mice using Image Pro Plus software (Media Cybernetics, Warrendale, PA, USA) driving a motorized stage on a BH2 Olympus microscope (Ciudad de México, México) with a 4× objective. Area sizes of the mossy fiber subfields were determined in 10 to 11 coronal sections per animal containing the DG. The volume of the mossy fiber subfields, the sum of areas measured, was multiplied by the inverse of the sampling fraction (6) and 40 (the section thickness in micrometers). For CNPase, the immunoreactivity was quantified in the hilus and GCL of the DG. Areas were outlined from the projected image, and sizes were determined using the area measurement followed by the optical density tool of the ImageJ software (v1.53t, NIH, Bethesda, MD, USA). For cellular and volume quantification, the analysis was conducted by a blinded analyzer on the experimental group.

### 4.6. Principal Component Analysis

We performed a principal component analysis (PCA) on physiological and behavioral measures to identify the responses to treatments. The PCA contemplated five behavioral and four physiological measurements across twelve mice samples. To determine the behavioral status, we included anxiety data with OF (open field) and EPM data (elevated plus maze), while depressed data were analyzed with FST values (forced swim test). Physiological data included Ki67, DCX, Calbindin, and CNPase. PCA was performed using the “factoextra” package [65,66]. All variables were scaled before PCA.

### 4.7. Statistical Analysis

Analyses were performed using Prism 5.0 (GraphPad). The results are presented as the mean ± standard error of the mean (SEM). We performed a one-way ANOVA for additional parameters, followed by the Bonferroni post-hoc test. All possible comparisons among the study groups were performed. In other cases, we performed a two-way ANOVA with the factors treatment (factor A) and time (factor B), followed by the Bonferroni post-hoc test. Differences were considered statistically significant at *p* ≤ 0.05. Also, a Kruskal–Wallis one-way analysis of variance on ranks followed by a Student–Newman–Keuls post hoc test was used.

## Figures and Tables

**Figure 1 ijms-24-13376-f001:**
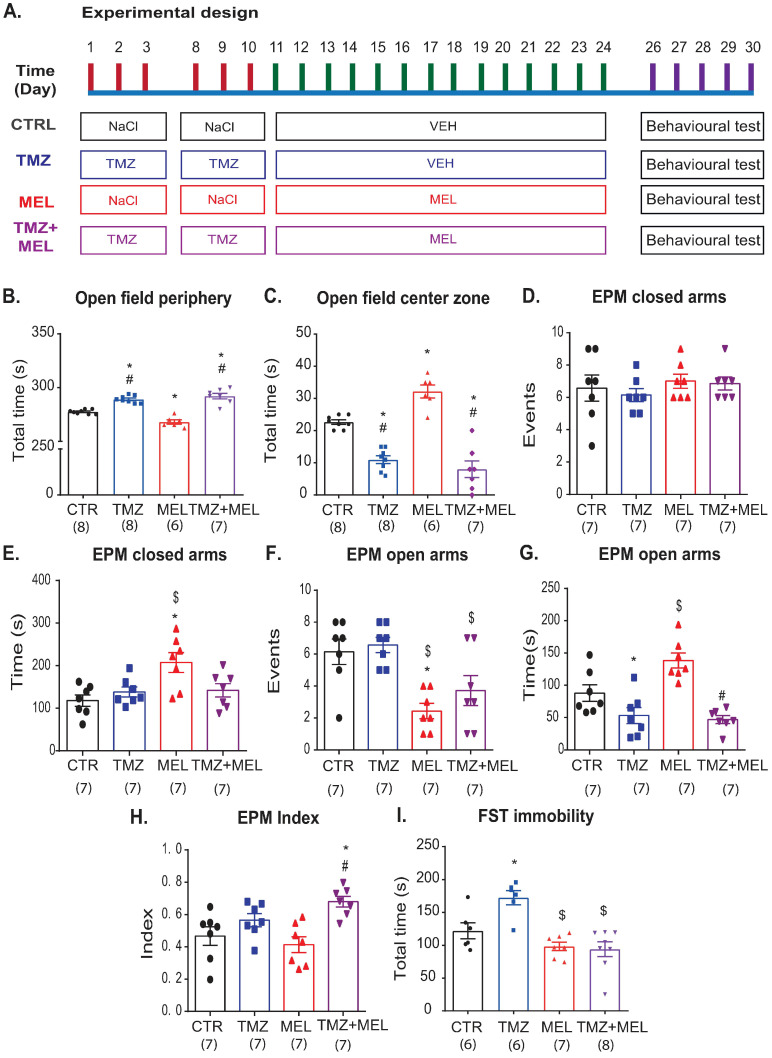
Temozolomide induced anxiety- and depression-like effects, but melatonin only prevented depression-like effects. (**A**) Experimental design. Female Balb/C mice received two cycles of temozolomide (TMZ: 25 mg/Kg) or saline solution (control, CTR: 0.9% NaCl) for three days (1–3 and 8–10 days) for two weeks. After the last administration of TMZ or saline solution, mice received melatonin (MEL: 8 mg/Kg) or the vehicle of MEL (VEH, ethanol: saline solution) for 14 days. Thus, the behavioral tests were performed from day 26 to 30. (**B**,**C**) Open field test. For five minutes, individual rodents were placed in a plexiglass box with clean sawdust on the floor. The time spent in the field’s periphery (**B**) or the center (**C**) was quantified. (**D**–**H**) Elevated plus maze. Mice were gently put at the center of the elevated plus maze face to open arms, and for five minutes we filmed the animal behavior to analyze the number of events in the closed (**D**) or open (**F**) arms and the time spent in the closed (**E**) or open (**G**) arms. (**H**) The anxiety index was calculated as indicated in the Materials and Methods section. (**I**) Forced swim test (FST). The immobility time in the forced swim test (FST) is shown. n = 6–8. Results were analyzed with a one-way ANOVA, followed by the Bonferroni post hoc test. But, when the normality test failed a Kruskal–Wallis one-way analysis of variance on ranks followed by a Student–Newman–Keuls post hoc test was applied (panel (**G**), open arms time). Differences were considered statistically significant at *p* ≤ 0.05. Significant differences are indicated with * used for those groups which are different from the control group (* vs. CTR group); # for those groups which are different from the melatonin group (# vs. MEL group); and $ for those groups which are different from the temozolomide group ($ vs. TMZ group).

**Figure 2 ijms-24-13376-f002:**
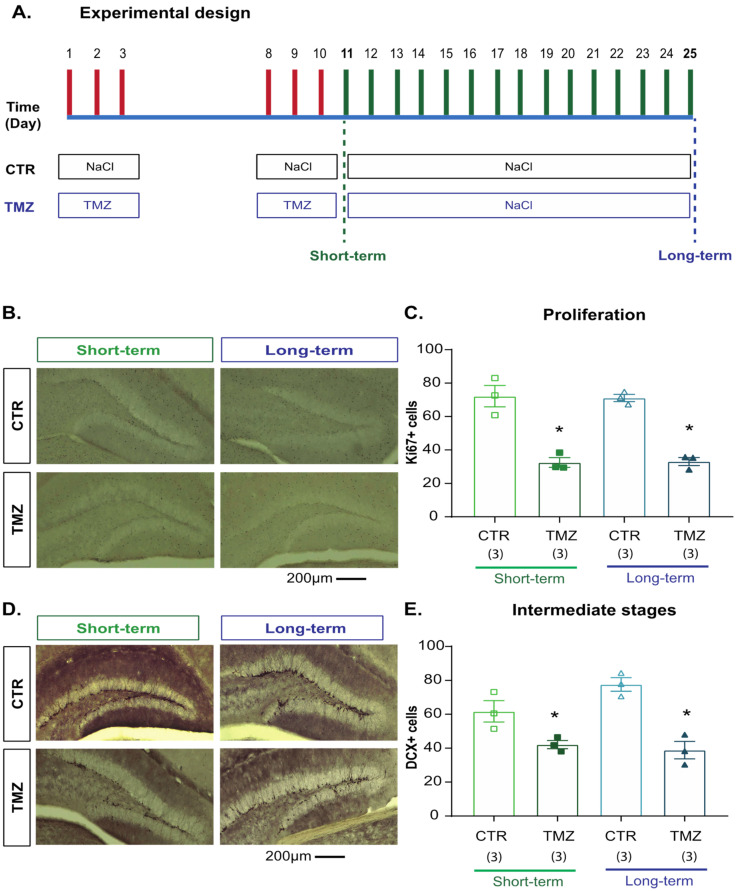
Temozolomide’s short- and long-term effects on cell proliferation and intermediate stages of hippocampal neurogenesis. (**A**) Experimental design. Female Balb/C mice received TMZ or saline solution on days 1–3 and 8–10. Mice were divided into control (CTR short-term and CTR long-term: N = 6) and TMZ (TMZ short-term, TMZ long-term: N = 6). For short-term evaluation, mice were euthanized at day 11, but for the long-term assessment mice were euthanized at day 25, and representative micrographs of Ki67 (**B**) or DCX (**D**) cells are shown. Scale bars 200 µm, respectively. Cumulative quantifications of Ki67- (**C**) or DCX-labeled cells (**E**) are shown. Results were analyzed with a two-way ANOVA, followed by the Bonferroni post hoc test. Factors were treatment and time of evaluation. Differences were considered statistically significant at *p* ≤ 0.05. n = 3. Significant differences are indicated with * for those groups different from the control group (* vs. CTR group) short- or long-term.

**Figure 3 ijms-24-13376-f003:**
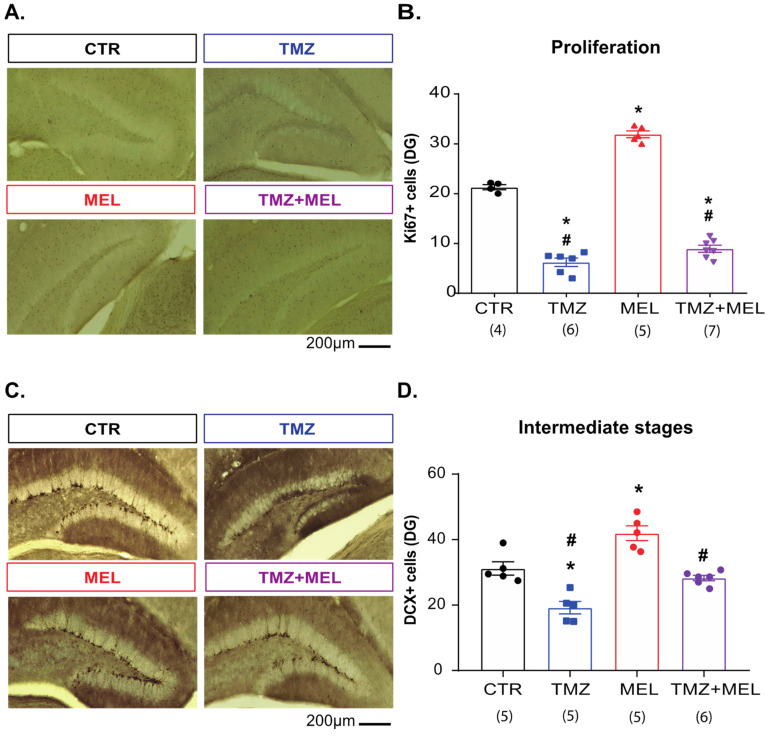
Melatonin prevents the effects of temozolomide on intermediate stages of hippocampal neurogenesis. Female Balb/C mice received temozolomide (TMZ) (25 mg/kg) or saline solution (0.9% NaCl) on days 1–3 and 8–10. Thus, mice were separated into four groups: control (CTR), (2) TMZ, (3) melatonin (MEL), and (4) TMZ + MEL; CTR and TMZ were injected intraperitoneally with saline solution from day 11 to 24. Mice of the MEL and TMZ + MEL groups were injected intraperitoneally with melatonin at the beginning of the dark phase of the light–dark cycle (8 mg/kg). Representative micrographs of Ki67 (**A**) or DCX (**C**) cells are shown. Scale bars 200 µm, respectively. The cumulative quantifications of Ki67 (**B**) or DCX (**D**) cells is shown. Results were analyzed with a one-way ANOVA, followed by the Bonferroni post hoc test. Differences were considered statistically significant at *p* ≤ 0.05. Differences were considered statistically significant at *p* ≤ 0.05. n = 4–7. Significant differences are indicated with * for those groups which are different from the control group (* vs. CTR group), # for those groups which are different from the melatonin group (# vs. MEL group).

**Figure 4 ijms-24-13376-f004:**
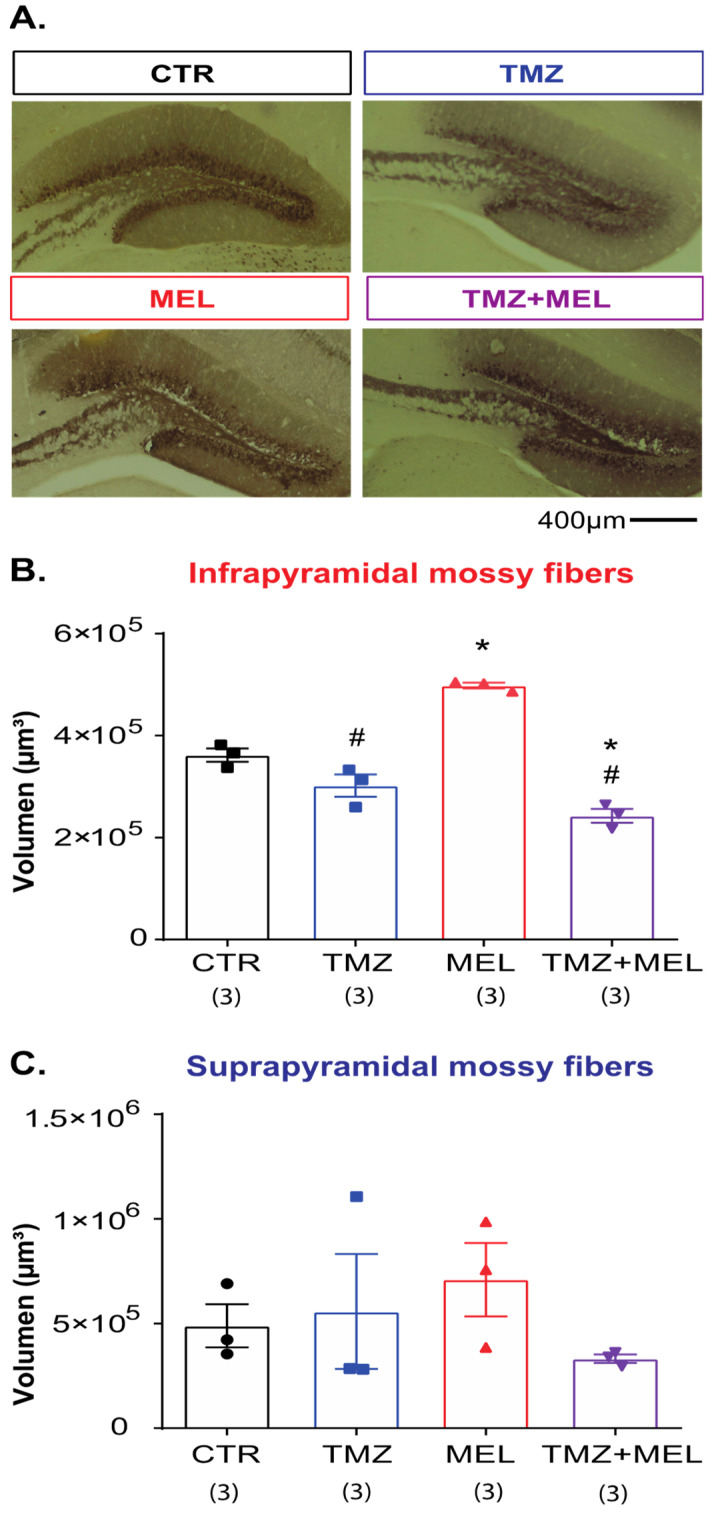
Melatonin increases the volume of the infrapyramidal mossy fiber tract. Female Balb/C mice received temozolomide (TMZ) (25 mg/kg) or saline solution (0.9% NaCl) on days 1–3 and 8–10. Thus, mice were separated into four groups: control (CTR), (2) TMZ, (3) melatonin (MEL), and (4) TMZ + MEL; CTR and TMZ were injected intraperitoneally with saline solution from day 11 to 24. Mice of the MEL and TMZ + MEL groups were injected intraperitoneally with MEL at the beginning of the dark phase of the light–dark cycle (8 mg/kg). Representative micrographs of calbindin (**A**) immunoreactivity are shown. Scale bar 400 µm, respectively. The volume of the infra- and suprapyramidal mossy fiber tract is shown in (**B**,**C**) as cubic micrometers. Results were analyzed with a one-way ANOVA, followed by the Bonferroni post hoc test. Differences were considered statistically significant at *p* ≤ 0.05. n = 3. Significant differences are indicated with * for those groups which are different from the control group (* vs. CTR group), # for those groups which are different from the melatonin group (# vs. MEL group).

**Figure 5 ijms-24-13376-f005:**
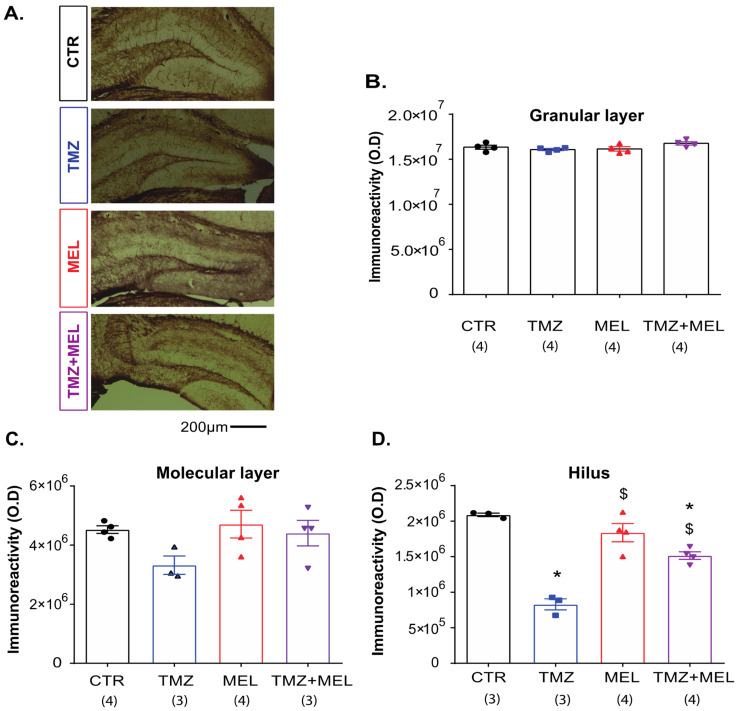
Melatonin prevented the decreased immunoreactivity of CNPase caused by temozolomide in the hilus of the dentate gyrus. Female Balb/C mice received temozolomide (TMZ) (25 mg/kg) or saline solution (0.9% NaCl) on days 1–3 and 8–10. Thus, mice were separated into four groups: control (CTR), (2) TMZ, (3) melatonin (MEL), and (4) TMZ + MEL; CTR and TMZ were injected intraperitoneally with saline solution from day 11 to 24. Mice of the MEL and TMZ + MEL groups were injected intraperitoneally with MEL at the beginning of the dark phase of the light–dark cycle (8 mg/kg). Representative micrographs of CNPase immunoreactivity are shown (**A**). Scale bar 200 µm, respectively. Immunoreactivity quantification of CNPase in the granular cell layer (GCL), the molecular layer, or in the hilus is shown (**B**–**D**) as optical density (O.D.). Results were analyzed with a one-way ANOVA, followed by the Bonferroni post hoc test. Differences were considered statistically significant at *p* ≤ 0.05. n = 3–4. Significant differences are indicated with * for all those groups different from the control group (* vs. CTR group), and $ indicates differences for all those groups different from the TMZ group ($ vs. TMZ group).

**Figure 6 ijms-24-13376-f006:**
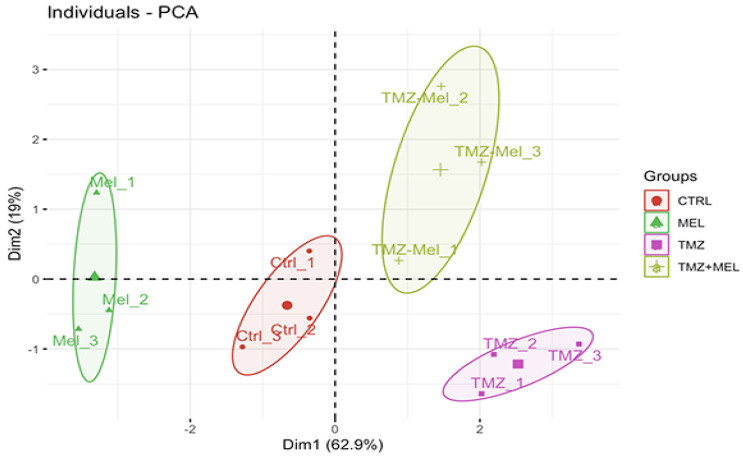
Principal component analysis. Principal component 1 (Dim1) versus 2 (Dim2) with individual dispersion are shown. Analysis included behavioral and physiological values using five behavioral and histological measurements across 12 samples and four treatments of mice divided into four groups: control (CTR), (2) TMZ, (3) melatonin (MEL), and (4) TMZ + MEL. The points correspond to each mouse’s PC1 and PC2 scores, and a 95% confidence interval defines the ellipses. Centroid ellipses are separated, indicating notable differences between treatments.

**Figure 7 ijms-24-13376-f007:**
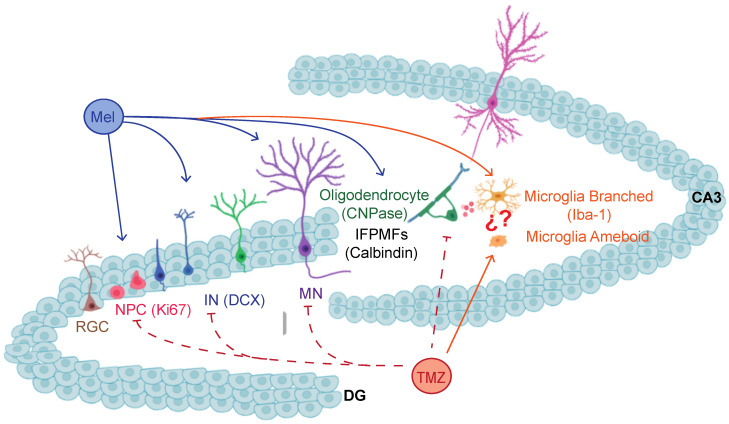
Model of the melatonin effects on structural dentate gyrus plasticity. The model suggests that melatonin (MEL) modulates several neurogenic events in the dentate gyrus (DG). Depending on the strain of mice, melatonin influences cell proliferation (Ki67), the intermediate stages based on the expression of doublecortin (DCX), and the increased volume of the infrapyramidal mossy fibers (Calbindin (IFPMFs)), which include the axons of new neurons projecting to the CA3. In mice treated with temozolomide (TMZ), melatonin prevents decreased Ki67 and DCX cells and the immunoreactivity for CNPase, a protein expressed in oligodendrocytes. The effects of melatonin may imply membrane receptor activation, and the regulation of cytoskeleton-associated proteins and signaling pathways. Illustration shows: RGC, radial glial cells; NPC, neural progenitor cells; IN, immature neuron; MN, mature neuron; IFPMFs, infrapyramidal mossy fibers; Mel, melatonin; TMZ: temozolomide.

**Table 1 ijms-24-13376-t001:** Eigenvalues.

Principal Component	Eigenvalues
**PC1**	**5.66**
**PC2**	**1.71**
PC3	0.79
PC4	0.42
PC5	0.20
PC6	0.135
PC7	0.045
PC8	0.015

Eigenvalues revealed two principal components (PC) with values higher than 1. PC1 and PC2 were considered for the principal component analysis.

**Table 2 ijms-24-13376-t002:** The explanation of the behaviors and physiology in PC1 and PC2 variations.

Measure	PC1 (%)	PC2 (%)
OF center time	**13.17**	10.07
OF periphery time	**13.17**	10.07
EPM open arms time	7.69	**16.68**
EPM close arms time	8.21	12.49
FST immobility time	4.62	**34.70**
Ki67-positive cells	**16.44**	0.16
DCX-positive cells	**12.62**	10.95
CB IFPMFT	**14.34**	2.76
CNPase immunoreactivity hilus	**9.70**	2.08

The Kaiser–Meyer–Olkin (KMO) revealed the appropriate use of the factorial analysis (0.51). Also, Bartlett’s test of homogeneity of variances was significant (BKs = 134.1, d.f. = 8, *p* < 2.2 × 10^−16^). Thus, variables loading higher for principal components 1 and 2 (PC1 or PC2) are indicated in bold.

## Data Availability

Data will be made available on request.

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
