# Peer review of "Melatonin Prevents Depression but Not Anxiety-like Behavior Produced by the Chemotherapeutic Agent Temozolomide: Implication of Doublecortin Cells and Hilar Oligodendrocytes"

_ijms, 2023, doi:10.3390/ijms241713376_

Round 1
Reviewer 1 Report
The manuscript proposes to verify if melatonin can reduce the anxiety and depression caused by TMZ, a molecule used in treating some brain cancers. The manuscript is generally very interesting, and its novelty is relevant. The manuscript’s structure is clear, and the results are concise. In view of improving the quality of the manuscript, I recommend that the authors verify some issues/questions I have during the evaluation of the manuscript:
- In section 2.2 of the results, the authors mentioned, “A recent study reported that TMZ induces depression-like behavior and alterations in the hippocampal neurogenic process [8]”. However, when I checked the reference, the article only analyzed the use of TMZ in a dose-intense approach in glioblastoma multiforme recurrent cases without giving any insight into depression-like behavior or the neurogenic process of hippocamp. Then, you should review this particular reference and consider adding in this sentence or in the discussion more information about how TMZ induces both effects you mentioned in the manuscript. You also present in the first sentence of the fourth paragraph of the discussion, which is not referenced at all.
- In section 2.3 of the results, you mentioned using Mel to reverse the effect of TMZ. The exposure to TMZ was of short or long-term? Clarify this, please.
- In section 2.4, still in the results, you wrote, “After we confirmed that MEL reverted the effects of TMZ on the number of DCX cells.” Since this study only evaluates 1 parameter regarding intermediate stages, and with this low number of animals, I think you should not use the word “confirmed that MEL reverted the effects…”. Instead, you can use “observed that MEL seems to revert the effects…”.
- Overall, the discussion seems confusing. The conclusions are a bit careless and bounce back and forth in the results in a confusing way. Specifically, the first three paragraphs of the discussion are extremely confusing and repetitive, and the information seems out of order.
- In the fourth paragraph, counting by the end of the discussion, you mention several targets and pathways that can explain the mechanisms observed (MT1/2, AKT, and others). I wonder if you have already tested some of the targets you presented as the explanation. If yes, I consider it relevant to present them here.
- Regarding section 4.1, there are several information I consider relevant to add, and also some questions I would like you to answer/include in the manuscript:
· Why did you use only female mice? And why have you chosen this particular mice strain?
· How do you calculate the number of mice needed for each experiment?
· To reduce the selection bias, did you randomize the animals in the treatment groups?
· To reduce detection bias, do you blind the outcome assessment?
· What kind of materials did you use for nesting and bedding? And how often did you change them?
· What are the cages used to maintain the animals? Are they calculated based on the size and weight of the animals?
· Did you use any enrichment material to reduce the animals’ stress? If yes, you should mention it.
Final thoughts:
Overall, the manuscript demonstrates that MEL has potential, but not for all the proposed ends. Nevertheless, I would like to ask if you consider it relevant to test the effect of MEL in mice with a brain tumor treated with TMZ because the tumor can lead to irreversible modifications in the tissue that cannot be observed in normal mice. Maybe exploring if MEL can produce the same results in those cases is relevant.

Author Response
POINT BY POINT RESPONSE
Melatonin prevented depression- but not anxiety-like behavior produced by the chemotherapeutic agent temozolomide: implication of hippocampal neurogenesis and hilar oligodendrocytes
ijms-2334769R1
REVIEWER 1
The manuscript proposes to verify if melatonin can reduce the anxiety and depression caused by TMZ, a molecule used in treating some brain cancers. The manuscript is generally very interesting, and its novelty is relevant. The manuscript’s structure is clear, and the results are concise. In view of improving the quality of the manuscript, I recommend that the authors verify some issues/questions I have during the evaluation of the manuscript:
Q1. In section 2.2 of the results, the authors mentioned, “A recent study reported that TMZ induces depression-like behavior and alterations in the hippocampal neurogenic process [8]”. However, when I checked the reference, the article only analyzed the use of TMZ in a dose-intense approach in glioblastoma multiforme recurrent cases without giving any insight into depression-like behavior or the neurogenic process of hippocamp. Then, you should review this particular reference and consider adding in this sentence or in the discussion more information about how TMZ induces both effects you mentioned in the manuscript. You also present in the first sentence of the fourth paragraph of the discussion, which is not referenced at all.
A1. We apologize for this mistake. In the new version of our manuscript, we carefully verified the references in the text. Also, we added some references to the manuscript.
Q2. In section 2.3 of the results, you mentioned using Mel to reverse the effect of TMZ. The exposure to TMZ was of short or long-term? Clarify this, please.
A2. Sorry for the lack of clarity. We modified the subheading from “Melatonin reverses effects of temozolomide on intermediate stages of hippocampal neurogenesis” to “Melatonin prevents long-term effects of temozolomide on intermediate stages of hippocampal neurogenesis.”
Q3 In section 2.4, still in the results, you wrote, “After we confirmed that MEL reverted the effects of TMZ on the number of DCX cells.” Since this study only evaluates 1 parameter regarding intermediate stages, and with this low number of animals, I think you should not use the word “confirmed that MEL reverted the effects…”. Instead, you can use “observed that MEL seems to revert the effects…”.
A3. We corrected it as this reviewer suggested us.
Q4 Overall, the discussion seems confusing. The conclusions are a bit careless and bounce back and forth in the results in a confusing way. Specifically, the first three paragraphs of the discussion are extremely confusing and repetitive, and the information seems out of order.
A4. We reordered the discussion section.
Q5 In the fourth paragraph, counting by the end of the discussion, you mention several targets and pathways that can explain the mechanisms observed (MT1/2, AKT, and others). I wonder if you have already tested some of the targets you presented as the explanation. If yes, I consider it relevant to present them here.
A5. At present, we have not explored the mechanism. However, we considered incorporating the information to infer a possible explanation of the results.
Q6 Regarding section 4.1, there are several information I consider relevant to add, and also some questions I would like you to answer/include in the manuscript:
Q6.1. Why did you use only female mice? And why have you chosen this particular mice strain?
A.6.1 All efforts were made to minimize animal suffering and reduce animal numbers. In this study, we used females instead of male mice due to the territorial effect present in males. In addition, female mice show less aggressive behavior than males after being randomly assigned to the experimental groups [62]. This strain of mice was also chosen due to its sensitivity to stress and its response to chronic antidepressants [63].
Q6.2. How do you calculate the number of mice needed for each experiment?
A6.2 We calculated the number of mice for each experiment with the G*Power software. The results were as follows:
F tests - ANOVA: Fixed effects, omnibus, one-way
Analysis: A priori: Compute required sample size
Input: Effect size f = 0.25
α err prob = 0.05
Power (1-β err prob) = 0.90
Number of groups = 4
Output: Noncentrality parameter λ = 14.5000000
Critical F = 2.6441945
Numerator df = 3
Denominator df = 228
Total sample size = 232
Sample size by group = 58
Actual power = 0.9018055
According to the IACUC observations, we needed to adjust the number of animals per group to minimize the number of mice per protocol. Thus, for behavior (experiment 1), we used between 7 to 8 mice per group. For the second experiment (experiment 2), in which we explored the effects of temozolomide on histology, we used 3 mice per group, in a total of 12 rodents. However, for histology done in brains derived from experiment 1, we used between 3 to 7 mice per group, depending on the resources available when we performed the histology. With the observations mentioned above, we recognize that the power of the statistics for every parameter was low according to the G*Power software. However, it was impossible to get access to 464 mice for the study. Nevertheless, we still found significant differences in some parameters evaluated here.
Q6.3. To reduce the selection bias, did you randomize the animals in the treatment groups?
A6.3. We apologize for the missing information. Animals were randomly assigned to the experimental group. This information was added to section 4.2 in the materials and methods.
Q6.4. To reduce detection bias, do you blind the outcome assessment?
A6.4. We apologize for the missing information. In the new version of our manuscript, we mention that the behavioral and histology were analyzed blind to the experimental group. Before the analysis was done, another researcher codified the videos and slices. Once the analysis was finished, the codes were revealed to reduce bias.
Q6.5. What kind of materials did you use for nesting and bedding? And how often did you change them?
A6.5. We added a piece of paper as nesting material. It was added twice a week.
Q.6.6 What are the cages used to maintain the animals? Are they calculated based on the size and weight of the animals?
A.6.6. The cages used for the animals have a size of 48x39x20 cm. The cages were chosen according to the official regulation of México to house experimental animals (NOM-062-ZOO-1999).
Q6.7. Did you use any enrichment material to reduce the animals’ stress? If yes, you should mention it.
A.6.7 No, we do not use tunnels or running wheels. We only used a piece of paper as nesting material.
Final thoughts: Overall, the manuscript demonstrates that MEL has potential, but not for all the proposed ends. Nevertheless, I would like to ask if you consider it relevant to test the effect of MEL in mice with a brain tumor treated with TMZ because the tumor can lead to irreversible modifications in the tissue that cannot be observed in normal mice. Maybe exploring if MEL can produce the same results in those cases is relevant.
AFT. We agree with the point of this reviewer. Our study demonstrates the potential of melatonin to prevent some of the long-term effects of temozolomide. However, it will be fascinating to study the effects of indole in an animal model of brain tumors. The discussion of the manuscript closed with the following paragraph: “Nevertheless, our study supports the relevance of MEL to revert the alterations caused by TMZ on behavior and neuroplasticity in female mice, pointing to the direction of the possible use of MEL to relieve the side effects of brain cancer treatment. However, this proposal must be probed in a preclinical study using cancer brain animal models, including female and male mice”.
Finally, we considered the necessity of renaming the manuscript to: Melatonin prevented depression- but not anxiety-like behavior produced by the chemotherapeutic agent temozolomide: implication of doublecortin cells and hilar oligodendrocytes, instead of the original name: Melatonin prevented depression- but not anxiety-like behavior produced by the chemotherapeutic agent temozolomide: implication of hippocampal neurogenesis and hilar oligodendrocytes. This change sticks to the parameters analyzed regarding the hippocampal neurogenic process.

Reviewer 2 Report
Paper titled (Melatonin reverses depression- but not anxiety-like behavior produced by the chemotherapeutic agent temozolomide: implication of hippocampal neurogenesis and hilar oligodendrocytes) by Cabrera-Muñoz tested the antidepressive effect of melatonin in mice subjected to temozolomide and documented non anxiogenic effect; authors claimed this effect is related to improving hippocampal neurogenesis and hilar oligodentrocytes. This study aim is NOT straight forward and conclusion sometimes do not cover the aim. I have the following serious comments which render this article hard to be acceptable.
1- Abstract: should be shortened & amended by some numerical values
2- Conclusion of abstract should be written directly : remove "brain cancer treatment" as the authors are trying to alleviate the drug adverse effects
3- Abstract conclusion: "However, it needs to be probed in a cancer model and with a different strain of mice and sex." this is not responding to the aim of the study!! authors should mention the mechanism by which melatonin afforded its effect.
4- Introduction: the first 5 lines are not necessary.
Make the intro concrete to focus on the target items
5- No need for P values <0.01, 0.001 , this does not mean your means are very different!, just mean SD values are small.
6- In resultts, separate the antidepressant effect from anxiolytic effect
7- Figure 1: what is FST?? explain in figure legend
8- Are the closed arms events or time important for testing anxiety?
9- Why the images are in black & white?
10- Figure 2 includes images and comparison for 2 study groups only, which is not a fair evidence in this situation
11- Ki67 is not a true evidence for neurogenesis, NeuroN is the correct one
12- Evidence is needed for data in Table 1
13- Discussion is too long and did not focus on the study results
14- What was the age and weight of mice at the begin of the study? how many mice in each group?
15- what was the source of the drug?
16- what was the ration of ethanol & saline in melatonin vehicle
-17- In methods: write and confirm that you highlighted every possible comparison between the study groups.
18- Were the animals trained on or Naive to the behavioral tasks? this is very important to be mentioned?
19- In page 16, how animals were euthanized? this is importent and worth mention
20- Mention at what data "When the normality test failed"
& in figure legends mention what was the test applied to each set of data
21- Methods in general lacks references at many occasions.
22- Use appropriate abbreviations for minutes, seconds...etc
23- Mention "n" in each illustation individually
24- Authors should give the source of chemicals, kits and antibodies completely and consistently (code, company, town, state and country) & version for software
Author Response
POINT BY POINT RESPONSE
Melatonin prevented depression- but not anxiety-like behavior produced by the chemotherapeutic agent temozolomide: implication of hippocampal neurogenesis and hilar oligodendrocytes
ijms-2334769R1
REVIEWER 2
Paper titled (Melatonin reverses depression- but not anxiety-like behavior produced by the chemotherapeutic agent temozolomide: implication of hippocampal neurogenesis and hilar oligodendrocytes) by Cabrera-Muñoz tested the antidepressive effect of melatonin in mice subjected to temozolomide and documented non anxiogenic effect; authors claimed this effect is related to improving hippocampal neurogenesis and hilar oligodentrocytes. This study aim is NOT straight forward and conclusion sometimes do not cover the aim. I have the following serious comments which render this article hard to be acceptable.
Q1.Abstract: should be shortened & amended by some numerical values.
A2. We included the increase or decrease percentages.
Q2. Conclusion of abstract should be written directly : remove "brain cancer treatment" as the authors are trying to alleviate the drug adverse effects
A2. We corrected it. The conclusion says: Thus, our results suggest that melatonin prevents the harmful effects of temozolomide by modulating doublecortin cells, hilar oligodendrocytes, and depression-like behavior. Our study could point out melatonin's beneficial effects for counteracting temozolomide's side effects.
Q3. Abstract conclusion: "However, it needs to be probed in a cancer model and with a different strain of mice and sex." this is not responding to the aim of the study!! authors should mention the mechanism by which melatonin afforded its effect.
A3. We corrected it. The conclusion says: Thus, our results suggest that Melatonin prevents the harmful effects of temozolomide by modulating doublecortin cells, hilar oligodendrocytes, and depression-like behavior. Our study could point out the beneficial effects of Melatonin in counteracting the side effects of temozolomide.
Q4. Introduction: the first 5 lines are not necessary.
Make the intro concrete to focus on the target items
A4. We eliminated the lines indicated by this reviewer. In the new version of the manuscript, the introduction starts with the next sentence: “Diagnosis and treatment are determining factors for the treatment of cancer [6-8]. In the case of brain cancer, some medications are better able to pass through the blood-brain barrier, such as Temozolomide (TMZ), an alkylating agent used with daily low-dose initially in combination with radiotherapy, followed by monthly doses of TMZ alone for long periods (6-12 months).”
Q5. No need for P values <0.01, 0.001 , this does not mean your means are very different!, just mean SD values are small.
A5. We agree with this reviewer that the p values <0.01 or 0.001 do not mean that means are very different between them. However, we are including the values because we describe two parameters in the groups. For instance: “Mice treated with TMZ showed more time in the periphery (Figure 1B; F3,26=12.95, p<0001) and lower time in the center (Figure 1C; F3,26=40.25, p<0001) than the control (p=0.017; p<0.001; respectively) and MEL (p<0.001; p<0.001; respectively) groups.”
Q6. In results, separate the antidepressant effect from anxiolytic effect
A6. We separated the results as this reviewer suggested us.
Q7. Figure 1: what is FST?? explain in figure legend
A7. We apologize for the missing information. In the figure legend, we wrote: “forced-swim test (FST).”
Q8. Are the closed arms events or time important for testing anxiety?
A8. Well, the behaviors assessed in the EPM are time spent (duration) in open and closed arms, the number of open and closed arms entries (events), and the total exploration (entries into all arms). The total exploration is calculated as the number of entries into any arm of the maze to distinguish between poor exploratory behavior, exploration limited to closed arms (avoidance), and free exploration. Some of the parameters mentioned above are used to calculate an anxiety index which is calculated as follows: Anxiety index = 1 – [(time spent in the open arms/total time on the maze)+(number of entries to the open arms/total exploration on the maze)/2], the anxiety index values range from 0 to 1 where an increase in the index expresses increased anxiety-like behavior (Cohen, H., Liu, T., Kozlovsky, N. et al. The Neuropeptide Y (NPY)-ergic System is Associated with Behavioral Resilience to Stress Exposure in an Animal Model of Post-Traumatic Stress Disorder. Neuropsychopharmacol 37, 350–363 (2012). https://doi.org/10.1038/npp.2011.230). In the original version of our manuscript, we did not include the anxiety index, but it was requested by reviewer 3. This information was incorporated in figure 1.
Q9. Why the images are in black & white?
A9. We apologize for it. In the new version of our manuscript, we incorporated representative images avoiding black and white. In the materials and methods section, we included more information about the process we followed to perform the
immunohistochemistry.
Q10. Figure 2 includes images and comparison for 2 study groups only, which is not a fair evidence in this situation
A7. We apologize for the lack of clarity in the description for experiment 2 in which we only analyzed the effect of temozolomide on Ki67- and doublecortin- positive cells at two-time points. In experiment 2, mice did not undergo behavioral tests as occurred in mice of experiment 1. From experiment 1, we are reporting behavior and histological results. In the latter, we include the same protein markers as in experiment 2. In the first version of our manuscript, we mentioned that: “A recent study reported that temozolomide induces depression-like behavior and alterations in the hippocampal neurogenic process. Thus, we analyzed the short- and long-term effect of TMZ on Ki67- and doublecortin-cells…”
Also, in section 2.4, we corrected the first sentence as follows: “Once we observed the short- and long-term effects of TMZ on Ki67- and DCX- cells, we analyzed whether MEL could prevent the long-term effects of TMZ on proliferative (Ki67)- and DCX cells in female Balb/C mice that underwent behavioral tests (Figure 3).”
Q11. Ki67 is not a true evidence for neurogenesis, NeuroN is the correct one
A11. We agree with the point of this reviewer. Ki67 is a protein expressed during the cell cycle of proliferative cells, and it cannot say that the generation of new neurons occurs in the dentate gyrus. Thus, this protein marker is studied to explore whether changes exist at the level of cell proliferation, an event of the neurogenic process that is the target of temozolomide. Along the manuscript, we corrected it for not using the term neurogenesis.
Q12. Evidence is needed for data in Table 1
A12. We thank this observation. In the new version of our manuscript, we included the next paragraph and tables related to the eigenvalues.
Finally, we performed the principal component analysis (PCA, Figure 6). We assembled a comprehensive set of five behavioral and four physiological parameters (Methods). The PCA revealed two main eigenvalues greater than 1 (Table 2). The PCA gave two main components explaining 81.9% variation (PC1 62.9%, PC2 19%) (Table 3; Figure 6). The PCA confirms the effect of treatment on behavioral and physiological values (Figure 6). Ki67-cells, infrapyramidal mossy fibers (CB IFPMF), time in the center of the open field (OF center time), time in the periphery of the open field (OF periphery time), and DCX- cells loaded strongly and concordantly on PC1 (62.9%, Table 3) indicating that PC1 primarily represents variation in anxiety behavior and proliferative and immature cells. PC2 (19%, Table 3) represents FST immobility time, EPM open arms time, and EPM close arms time; this may indicate a change in behavior with the TMZ-Mel treatment. The Kaiser-Meyer-Olkin (KMO) revealed the appropriate use of the factorial analysis (0.51). Also, Bartlett’s test of homogeneity of variances was significant (BKs=134.1, d.f.=8, p<2.2x10-16), suggesting a significant and relevant correlation between the variables and the factor model.
Q13. Discussion is too long and did not focus on the study results.
A13. We thank this observation because it allowed us to reorganize the discussion section.
Q14. What was the age and weight of mice at the begin of the study? how many mice in each group?
A14. We included the missing information in the materials and methods section.
Q15. what was the source of the drug?
A15. We included the missing information in the materials and methods section.
Q16. what was the ration of ethanol & saline in melatonin vehicle
A16. We included the missing information: “Melatonin (Sigma, Naucalpan, Estado de México, México) was freshly prepared daily and dissolved in a minimum volume of pure ethanol plus saline solution (0.9% NaCl) with the final working concentration of 8 mg/kg body weight. The final volume of ethanol in the melatonin vehicle was less than 1% in a ratio of 1:100 (ethanol:saline solution).”
Q17. In methods: write and confirm that you highlighted every possible comparison between the study groups.
A17. In the materials and methods section (4.7), we included the phrase: “All possible comparisons among the study groups were performed.”
Q18. Were the animals trained on or Naive to the behavioral tasks? this is very important to be mentioned?
A18. Animals were naïve to the behavioral tests. In the materials and methods section (4.3), we included the phrase: “animals were naïve to the behavioral tests.”
Q19. In page 16, how animals were euthanized? this is importent and worth mention
A19. Mice were euthanized by decapitation two hours after the behavioral test. This information was included in the new version of our manuscript.
Q20. Mention at what data "When the normality test failed" and in figure legends mention what was the test applied to each set of data
A20. We apologize for the inclusion of the phrase mentioned above by mistake. We have corrected the text, and in the figure legends, we mention the test applied to each data set.
Q21. Methods in general lacks references at many occasions.
A21. We corrected it.
Q22. Use appropriate abbreviations for minutes, seconds...etc
A22. We corrected it.
Q23. Mention "n" in each illustration individually
A23. We corrected it.
Q24. Authors should give the source of chemicals, kits and antibodies completely and consistently (code, company, town, state and country) & version for software
A24. We corrected it.
Finally, we considered the necessity of renaming the manuscript to: Melatonin prevented depression- but not anxiety-like behavior produced by the chemotherapeutic agent temozolomide: implication of doublecortin cells and hilar oligodendrocytes, instead of the original name: Melatonin prevented depression- but not anxiety-like behavior produced by the chemotherapeutic agent temozolomide: implication of hippocampal neurogenesis and hilar oligodendrocytes. This change sticks to the parameters analyzed regarding the hippocampal neurogenic process.

Reviewer 3 Report
The authors investigated the effect of MLT in reversing anxiety- and depression-like behaviours and neurogenesis indiced by the anti-cancer drug temozolomide
(TMZ). MLT reversed the immobility in the FST, but not the the total time spent in the central area in the OFT or in the open arm in the EPM test. Histological findings also showed that TMZ produced short- and long-term decrements in cell proliferation (quantified as Ki67 positive cells) and intermediate stages of the neurogenic process (doublecortin + cells). However, Melatonin reversed the effects of TMZ with the increased number of doublecortin-positive cells and the immunoreactivity of 2'3'-Cyclic-nucleotide-3'phosphodiesterase (CNPase), a marker for oligodendrocytes, in the hilar portion of the dentate gyrus.
Abstract:
- the introduction about cancer incidence can be shorted as this is not the main focus of the ms.
Results:
- In order to summarize the findings from the EPM. authors can calculate the anxiety index (Cohen et al., 2008).
- Authors should briefly mention here why they focus on the supra- and infra-pyramidal mossy fibers, and which biological roles these fibers play in the neurogenesis and depression.
- Fig. 5: the scale in the y axis is confusing, particularly in B and C. Please fix it.
Discussion:
Authors should clearly state here that one of the limitation of this study is the fact that they are using a murine cancer model which involves TMZ as treatment. Also, they study should be conducted in male mice.
Minor:
- FST has been long time considered a test to measure depressive-live behavior. To confirm and strength the findings about anxiety and depressive-like behavior, the Novelty-suppressed feeding (a paradigm used to assess anxiety-like behavior but also sensitive to chronic antidepressant treatment) and sucrose preference test could be also used to measure anedonia-like behavior.
Author Response
POINT BY POINT RESPONSE
Melatonin prevented depression- but not anxiety-like behavior produced by the chemotherapeutic agent temozolomide: implication of hippocampal neurogenesis and hilar oligodendrocytes
ijms-2334769R1
REVIEWER 3
The authors investigated the effect of MLT in reversing anxiety- and depression-like behaviours and neurogenesis indiced by the anti-cancer drug temozolomide (TMZ). MLT reversed the immobility in the FST, but not the total time spent in the central area in the OFT or in the open arm in the EPM test. Histological findings also showed that TMZ produced short- and long-term decrements in cell proliferation (quantified as Ki67 positive cells) and intermediate stages of the neurogenic process (doublecortin + cells). However, Melatonin reversed the effects of TMZ with the increased number of doublecortin-positive cells and the immunoreactivity of 2'3'-Cyclic-nucleotide 3'phosphodiesterase (CNPase), a marker for oligodendrocytes, in the hilar portion of the dentate gyrus.
Q1. Abstract:
- the introduction about cancer incidence can be shorted as this is not the main focus of the ms.
A1. We corrected the abstract as indicated by the reviewers.
Q2. Results:
Q2.1. In order to summarize the findings from the EPM. authors can calculate the anxiety index (Cohen et al., 2008).
A2. We thank this observation. In the new version of figure 1, we included the anxiety index.
Q2.2. Authors should briefly mention here why they focus on the supra- and infra-pyramidal mossy fibers, and which biological roles these fibers play in the neurogenesis and depression.
A2.2 It is true. We omitted to mention the previous studies related to axons of the newborn neurons and mossy fibers. Also, we omitted to mention our previous study, in which we showed that melatonin increases the volume of mossy fibers. In the new version of our manuscript, both references were added in the results section. Also, in the discussion section, we incorporated the relevance of the mossy fibers for neurogenesis and depression.
Q2.3. Fig. 5: the scale in the y axis is confusing, particularly in B and C. Please fix it.
A2.3 We corrected it. In the new histograms, the y axis starts from 0.
Q3.Discussion: Authors should clearly state here that one of the limitation of this study is the fact that they are using a murine cancer model which involves TMZ as treatment. Also, they study should be conducted in male mice.
A3. In the new version of our manuscript, we clearly state the limitations indicated here. The last paragraph of the discussion says: “Nevertheless, our study supports the relevance of MEL to revert the alterations caused by TMZ on behavior and neuroplasticity in female mice, pointing to the direction of the possible use of MEL to relieve the side effects of brain cancer treatment. However, a limitation of our study is that we did not use a murine cancer model, and our results need to be replicated in a preclinical study using cancer brain animal models, including female and male mice.
Minor:
FST has been long time considered a test to measure depressive-live behavior. To confirm and strength the findings about anxiety and depressive-like behavior, the Novelty-suppressed feeding (a paradigm used to assess anxiety-like behavior but also sensitive to chronic antidepressant treatment) and sucrose preference test could be also used to measure anhedonia-like behavior.
A4. This observation is very interesting. In previous studies performed by our group, we have used a chronic mild stress model to confirm the effects of melatonin on depression and some aspects of the neurogenic process (i.e., Vega-Rivera et al. 2021). Unfortunately, we did not include an anhedonia-behavioral test or novelty-suppressed feeding for the present study. To perform those additional behavioral tests, getting authorization for the IACUC will be necessary, which will take us a long time to resubmit the manuscript. However, if this experiment is mandatory, we can make the effort.
Finally, we considered the necessity of renaming the manuscript to: Melatonin prevented depression- but not anxiety-like behavior produced by the chemotherapeutic agent temozolomide: implication of doublecortin cells and hilar oligodendrocytes, instead of the original name: Melatonin prevented depression- but not anxiety-like behavior produced by the chemotherapeutic agent temozolomide: implication of hippocampal neurogenesis and hilar oligodendrocytes. This change sticks to the parameters analyzed regarding the hippocampal neurogenic process.

Round 2
Reviewer 2 Report
The revised version of paper titled (Melatonin reverses depression- but not anxiety-like behavior produced by the chemotherapeutic agent temozolomide: implication of hippocampal neurogenesis and hilar oligodendrocytes) was NOT improved compared to the original one, the study design actually and methodologies are very deficent that cannot be improved.
Authors did not reply adequately to the previous comments and most of answers are not relevant to the question or INSUFFECIENT.
Beside:
Using alcohol as a vehicle in anxiety study is NOT correct due to its anxiolytic action.
Comments to statisatcial analysis are fatal and NOT corrected & hence cannot rely on these data. Authors have to test normality of data distribution and perform two-way ANOVA.
Iam sorry to say that the original images in black & white ARE NOT the same like the revised colored images. !!!!
How authors performed image analysis to these low power images? this is not indicative to the microscopic fields. Images are of poor resolution. Again, they are NOT the original images.
Author Response
POINT BY POINT RESPONSE
Melatonin prevented depression- but not anxiety-like behavior produced by the chemotherapeutic agent temozolomide: implication of hippocampal neurogenesis and hilar oligodendrocytes
ijms-2334769R2
REVIEWER 2
Reviewer observations appear in blue color
Responses to the first-round observations appear in yellow color
Responses to the second-round observations appear in green color
FIRST-ROUND OBSERVATIONS AND RESPONSES
Paper titled (Melatonin reverses depression- but not anxiety-like behavior produced by the chemotherapeutic agent temozolomide: implication of hippocampal neurogenesis and hilar oligodendrocytes) by Cabrera-Muñoz tested the antidepressive effect of melatonin in mice subjected to temozolomide and documented non anxiogenic effect; authors claimed this effect is related to improving hippocampal neurogenesis and hilar oligodentrocytes. This study aim is NOT straight forward and conclusion sometimes do not cover the aim. I have the following serious comments which render this article hard to be acceptable.
SECOND-ROUND OBSERVATIONS AND RESPONSES
The revised version of paper titled (Melatonin reverses depression- but not anxiety-like behavior produced by the chemotherapeutic agent temozolomide: implication of hippocampal neurogenesis and hilar oligodendrocytes) was NOT improved compared to the original one, the study design actually and methodologies are very deficent that cannot be improved.
Authors did not reply adequately to the previous comments and most of answers are not relevant to the question or INSUFFECIENT.
Beside:
Using alcohol as a vehicle in anxiety study is NOT correct due to its anxiolytic action.
Comments to statisatcial analysis are fatal and NOT corrected & hence cannot rely on these data. Authors have to test normality of data distribution and perform two-way ANOVA.
Iam sorry to say that the original images in black & white ARE NOT the same like the revised colored images. !!!!
How authors performed image analysis to these low power images? this is not indicative to the microscopic fields. Images are of poor resolution. Again, they are NOT the original images.
ABSTRACT
Q1. Abstract: should be shortened & amended by some numerical values.
*ANSWER FIRST-ROUND OF REVIEW
A1. In the first round, we added the increase or decrease percentages of the parameters here evaluated. We hope that these are the numerical values required by this reviewer.
Q2. Conclusion of abstract should be written directly, remove "brain cancer treatment" as the authors are trying to alleviate the drug adverse effects.
*ANSWER FIRST-ROUND OF REVIEW
A2. We corrected it. The conclusion says: Thus, our results suggest that melatonin prevents the harmful effects of temozolomide by modulating doublecortin cells, hilar oligodendrocytes, and depression-like behavior. Our study could point out melatonin's beneficial effects for counteracting temozolomide's side effects.
Q3. Abstract conclusion: "However, it needs to be probed in a cancer model and with a different strain of mice and sex." this is not responding to the aim of the study!! authors should mention the mechanism by which melatonin afforded its effect.
*ANSWER FIRST-ROUND OF REVIEW
A3. We corrected it. The conclusion says: Thus, our results suggest that Melatonin prevents the harmful effects of temozolomide by modulating doublecortin cells, hilar oligodendrocytes, and depression-like behavior. Our study could point out the beneficial effects of Melatonin in counteracting the side effects of temozolomide. In the first round, we included the information regarding to the possible mechanism which may underlay the benefits of melatonin to prevent the effects of temozolomide.
**ANSWER SECOND-ROUND OF REVIEW
To clarify the point of this reviewer in the first revision we are including the abstract to highlight the substantial modifications.
Melatonin is a hormone synthesized by the pineal gland with neuroprotective and neurodevelopmental effects. Also, melatonin acts as an antidepressant by modulating the generation of new neurons in the dentate gyrus of the hippocampus. The positive effects of melatonin on behavior and neural development may suggest it is used for reverting stress but also on the alterations produced by chemotherapeutic drugs influencing behavior and brain plasticity. In this sense, temozolomide, an alkylating/anti-proliferating agent used in treating brain cancer, is associated with decreased cognitive functions and depression. We hypothesized that melatonin might prevent the effects of temozolomide on depression- and anxiety-like behavior by modulating some aspects of the neurogenic process in adult Balb/C mice. Mice were treated with temozolomide (25 mg/kg) for three days of two weeks, followed by melatonin (8 mg/kg) for 14 days. Temozolomide produced short- and long-term decrements in cell proliferation (Ki67 positive cells: 54.89%, 53.38%; respectively) and intermediate stages of the neurogenic process (doublecortin positive cells: 68.23%, 50.08%; respectively). However, melatonin prevented the long-term effects of temozolomide with the increased number of doublecortin-positive cells (47.21%) and the immunoreactivity of 2' 3'-Cyclic-nucleotide-3 phosphodiesterase (CNPase: 82.66%), an enzyme expressed by mature oligodendrocytes, in the hilar portion of the dentate gyrus. The effects of melatonin in the temozolomide group occurred with decreased immobility in the forced swim test (45.55%) but not anxiety-like behavior. Thus, our results suggest that melatonin prevents the harmful effects of temozolomide by modulating doublecortin cells, hilar oligodendrocytes, and depression-like behavior tested in the forced swim test. Our study could point out melatonin's beneficial effects for counteracting temozolomide's side effects.
INTRODUCTION
Q4. Introduction: the first 5 lines are not necessary.
Make the intro concrete to focus on the target items
*ANSWER FIRST-ROUND OF REVIEW
A4. In the first round, we eliminated the lines indicated by this reviewer making concrete the introduction by including the relevant information for the present study. Now the first paragraph says:
“Diagnosis and treatment are determining factors for the treatment of cancer [6-8]. In the case of brain cancer, some medications are better able to pass through the blood-brain barrier, such as temozolomide (TMZ), an alkylating agent used with daily low-dose initially in combination with radiotherapy, followed by monthly doses of TMZ alone for long periods (6-12 months) [1-7]. TMZ is indicated for glioblastoma multiforme, the most frequent primary brain cancer with a higher incidence in men than women. Although secondary glioblastoma tumors, which progress from low-grade diffuse astrocytoma or anaplastic astrocytoma, are seen more frequently in women [1-5]. TMZ shows promising efficacy in treating brain cancer, but patients treated with TMZ report depressive episodes and progressive decreased cognitive abilities [1-7]. Recently, in C57Bl/6J mice, it was mimicked the cyclic treatment administered in the clinic with daily injections of TMZ on three consecutive days every week causing diminished the number of new neurons in the dentate gyrus (DG) of the hippocampus, and this decrease was associated with depressive-like behavior [8]. “
RESULTS
Q5. No need for P values <0.01, 0.001, this does not mean your means are very different!, just mean SD values are small
*ANSWER FIRST-ROUND OF REVIEW
**ANSWER SECOND-ROUND OF REVIEW
A5. In the first round, we mentioned the why we conserved both values.
We agree with this reviewer that the p values <0.01 or 0.001 do not mean that means are very different between them. However, we are including both values because we describe two different parameters in the groups with specific p values. In the text, the information says: “Mice treated with TMZ showed more time in the periphery (Figure 1B; F3,26=12.95, p<0001) and lower time in the center (Figure 1C; F3,26=40.25, p<0001) than the control (p=0.017; p<0.001; respectively) and MEL (p<0.001 in both parameters; respectively) groups.”
**ANSWER SECOND ROUND OF REVIEW
The reason to conserve both p values was that we described two different parameters with real and specific p values. Also, these results derived from an independent one-way ANOVA because at least in this case we cannot applied a two-way ANOVA due that we do not have two factors.
Q6. In results, separate the antidepressant effect from anxiolytic effect.
*ANSWER FIRST ROUND OF REVIEW
A6. We separated the results as this reviewer suggested us. In the new version of our manuscript, you can read as follows:
**ANSWER SECOND ROUND OF REVIEW
FOR THIS ROUND OF REVIEW, WE INCLUDED THE INFORMATION THAT IS INCLUDED IN THE MANUSCRIPT.
2.1 Temozolomide-induced anxiety-like behavior, which melatonin does not prevent.
First, we explored the long-term effects of TMZ on behavior (Figure 1A). Fifteen days after the last administration of TMZ, mice were exposed to the open field test (Figure 1 B, C). Mice treated with TMZ showed more time in the periphery (Figure 1B; F3,26=12.95, p<0001) and lower time in the center (Figure 1C; F3,26=40.25, p<0001) than the control (p=0.017; p<0.001; respectively) and MEL (p<0.001 for both parameters, respectively) groups. Mice that received MEL after TMZ did not show modifications in the time in the periphery (p=1) or the center (p=1) compared with the TMZ group.
Also, mice were tested in the elevated plus maze (EPM). The EPM test (Figure 1D-G; Closed arms (events): F3,27=0.48, p=0.69; Closed arms (time): F3,27=5.45, p=0.005; Opened arms (events): F3,27=7.83, p<0.001; Opened arms (time): H=17.29, d.f.=3, p<0.001) revealed that TMZ compared with the control group, did not modify the number of events (Figure 1D, p=0.49) or the time in the closed arms (Figure 1E, p=1). Similar effects of TMZ were seen in the number of events in the open arms (p=1; Figure 1F), but the time spent in the open arms was lower than in the control group (q=4.78, p<0.05; Figure 1G), whereas MEL treated mice showed increased time spent in the opened arms (q=4.9, p<0.05) and decreased events in the opened arms (Figure 1F; p=0.006) compared with the control group. But the time spent in the open arms was higher in the MEL-treated group compared with the TMZ (q=6.12, p<0.05, Figure 1G) or in TMZ followed by MEL (q=5.23, p<0.05). Also, the number of events in the open arms was higher in the TMZ than in the MEL group (p=0.002, Figure 1F), and it showed a trend toward a significant difference (p=0.052) compared with mice that received TMZ followed by MEL. Mice treated with TMZ did not show differences in the number of events in the closed arms compared with the other groups (Figure 1D). Also, the analysis of the time spent in the closed arms (Figure 1E) showed similar results among groups, although MEL significantly increased the time spent in the closed arms than TMZ and control (p=0.042, p=0.005; respectively). Interestingly, TMZ and TMZ followed by MEL, did not show differences (p=1). Thus, we calculated the anxiety index (AI) (Figure 1H; F3,27=6.48, p=0.002). The AI confirmed that melatonin could not prevent the anxiety-like behavior produced by TMZ (p=0.003). These results suggest that MEL does not prevent anxiety-like behavior caused by TMZ.
2.2 Temozolomide-induced depression-like effects, which melatonin prevents.
Also, we evaluated the effects of TMZ on despair-like behavior in the forced-swim test (FST) (Figure 1H). TMZ increased the immobility time in the FST (Figure 1H; F3,26=11.54, p<0.001) compared with MEL (p<0.001), with MEL after TMZ treatment (p<0.001) or with the control group (p=0.029); respectively. These results suggest that MEL prevented to depression-like behavior but not anxiety-like behavior in the open field test caused by TMZ.
Q7. Figure 1: what is FST?? explain in figure legend
*ANSWER FIRST ROUND OF REVIEW
A7. We apologize for the missing information. In the figure legend, we wrote: “forced-swim test (FST).” Please see below the full figure legend.
**ANSWER SECOND ROUND OF REVIEW
FOR THIS ROUND OF REVIEW, WE INCLUDED THE INFORMATION THAT IS INCLUDED IN THE MANUSCRIPT.
Figure 1. Temozolomide induced anxiety- and depression-like effects, but melatonin only reverted depression-like effects. A) Experimental design, female Balb/C mice received two cycles of temozolomide (TMZ; 25 mg/Kg) or saline solution (control, CTR; 0.9% NaCl) for three days (1-3 and 8-10 days) for two weeks. After the last administration of TMZ or saline solution, mice received melatonin (MEL; 8 mg/Kg) or the vehicle of MEL (VEH, ethanol: saline solution) for 14 days. Thus, the behavioral tests were performed from day 26 to 30. B-C) Open field test. Individual rodents were placed in a plexiglass box with clean sawdust on the floor for five minutes. The time spent in the field's periphery (B) or the center (C) was quantified. D-G) Elevated plus maze. Mice were gently put at the center of the elevated plus maze face to open arms, and for five minutes, we filmed the animal behavior to analyze the number of events in the closed (D) or open (F) arms and the time spent in the closed (E) or open (G). H) The anxiety index was calculated as indicated in the materials and methods section. I) Forced swim test (FST). The immobility time is shown. n=7–8. Significant differences are indicated with * used for those groups which are different from control (*vs. CTR group); # for those groups which are different from melatonin (# vs. MEL group); $ for those groups which are different from temozolomide ($ vs. TMZ group). Results were analyzed with a one-way ANOVA followed by the Bonferroni post hoc test. But when the normality test failed a Kruskal-Wallis one-way analysis of variance on ranks followed by Student-Newman-Keuls post hoc was applied (Open arms time). Differences were considered statistically significant at p≤0.05.
Q8. Are the closed arms events or time important for testing anxiety?
*ANSWER FIRST ROUND OF REVIEW
**ANSWER SECOND ROUND OF REVIEW
A8. Well, most of the studies report the number of events and time spent in the open arms. However, for the anxiety index, it is necessary to know the number and time spent in the closed arms. Thus, we considered that reporting all the parameters of the EPM is relevant to understand the effects on anxiety. Then, behaviors assessed in the EPM are time spent (duration) in open and closed arms, the number of open and closed arms entries (events), and the total exploration (entries into all arms). The total exploration is calculated as the number of entries into any arm of the maze to distinguish between poor exploratory behavior, exploration limited to closed arms (avoidance), and free exploration. Some of the parameters mentioned above are used to calculate an anxiety index which is calculated as follows: Anxiety index = 1 – [(time spent in the open arms/total time on the maze)+(number of entries to the open arms/total exploration on the maze)/2], the anxiety index values range from 0 to 1 where an increase in the index expresses increased anxiety-like behavior (Cohen, H., Liu, T., Kozlovsky, N. et al. The Neuropeptide Y (NPY)-ergic System is Associated with Behavioral Resilience to Stress Exposure in an Animal Model of Post-Traumatic Stress Disorder. Neuropsychopharmacol 37, 350–363 (2012). https://doi.org/10.1038/npp.2011.230).
In the original version of our manuscript, we did not include the anxiety index, but it was requested by reviewer 3. This information was incorporated in figure 1.
Q9. Why the images are in black & white?
*ANSWER FIRST ROUND OF REVIEW
A9. The answer to this observation done in the first revision was: “We apologize for it. In the new version of our manuscript, we incorporated representative images avoiding black and white. In the materials and methods section, we included more information about the process we followed to perform the immunohistochemistry.”
Q9.1 About the new revision (R1) this reviewer commented that:
“I am sorry to say that the original images in black & white ARE NOT the same like the revised colored images. !!!!”
“How authors performed image analysis to these low power images? this is not indicative to the microscopic fields. Images are of poor resolution. Again, they are NOT the original images.”
**ANSWER SECOND ROUND OF REVIEW
A9.1 “It is true. We incorporated better images than the previous images included in the original version. New representative images are in their original format without changing the mode of the images to grayscale as the ones in the original version.
The images were captured in a high-quality format. However, the figure files included in the manuscript and the pdf version were in the jpeg format, affecting our figures' quality. In the new version, we included images in a higher resolution.
Q10. Figure 2 includes images and comparison for 2 study groups only, which is not a fair evidence in this situation
*ANSWER FIRST ROUND OF REVIEW
**ANSWER SECOND ROUND OF REVIEW
A7. We apologize for the lack of clarity in the description for experiment 2 in which we only analyzed the effect of temozolomide on Ki67- and doublecortin- positive cells at two-time points (short-term and long-term). In experiment 2, mice did not undergo behavioral tests as occurred in mice of experiment 1.
In the section 2.3, we mentioned the next information:
“A recent study reported that TMZ induces depression-like behavior and alterations in the hippocampal neurogenic process [8]. Thus, we analyzed the short- and long-term effects of TMZ on Ki67- and DCX- cells in the subgranular- and granular-cell layers in the DG of the hippocampus, respectively (Figure 2) in rodents that did not go to the behavioral tests. Cellular quantifications were performed the day after the last administration of TMZ or 14 days after the TMZ treatment finished (Figure 2A).”
This information supports the why we performed the experiment 2 including two study groups but analyzed at different time points in different groups of mice. For that reason, we applied a two-way ANOVA, as is indicated in the manuscript, figure legend and in the statistical analysis section included in materials and methods.
Please see below the information that we included in the first revision:
“Cellular quantifications were performed the day after the last administration of TMZ or 14 days after the TMZ treatment finished (Figure 2A). Two-way ANOVA for Ki67 cells showed the main effect of treatment (F3,8=41.265, p <0.001) but not at the time of evaluation (F3,8=0.569, p=0.47). Thus, proliferative cells in mice treated with TMZ decreased at both assessment time points (Figure 2B, C). Similar effects were seen after the quantification of DCX cells; TMZ produced a short- and long-term decreased number of DCX cells (main effect of treatment (F3.8=4.96, p =0.040; Figure 2D, E).”
Also, the figure legend was modified as follows:
Figure 2. Temozolomide's short- and long-term effects on cell proliferation and intermediate stages of hippocampal neurogenesis. A) Experimental design; female Balb/C mice received TMZ or saline solution on days 1-3 and 8-10. Mice were divided into control (CTR short-term and CTR long-term; N=6) and TMZ (TMZ short-term, TMZ long-term; n=6). For short-term evaluation, mice were euthanized at day 11, but for the long-term assessment, mice were euthanized at day 25, and representative micrographs of Ki67- (B) or DCX- (D cells are shown. Scale bars 25 µm and 200 µm, respectively. Cumulative quantification of Ki67 (E) or DCX-labeled cells (E) is shown. Significant differences are indicated with * for those groups different from control (* vs. CTR) short or long-term. Results were analyzed with a two-way ANOVA followed by the Bonferroni post hoc test. Differences were considered statistically significant at p≤0.05. n=3.
Q11. Ki67 is not true evidence for neurogenesis, NeuroN is the correct one
*ANSWER FIRST-ROUND OF REVIEW
A11. The response that we gave to this observation in the first round was: “We agree with the point of this reviewer. Ki67 is a protein expressed during the cell cycle of proliferative cells, and it cannot say that the generation of new neurons occurs in the dentate gyrus. Thus, this protein marker is studied to explore whether changes exist at the level of cell proliferation, an event of the neurogenic process that is the target of temozolomide. Along the manuscript, we corrected it for not using the term neurogenesis.”
**ANSWER SECOND-ROUND OF REVIEW
To extend our answer to this observation we want to mention that “Ki67 (used for cell proliferation) and NeuN (mature neurons) protein markers by separated are not enough to conclude effects of treatments on neurogenesis. To reach the conclusion of effects of pharmacological interventions on hippocampal neurogenesis, it is necessary to performed analysis on double-labeling cells (BrdU/NeuN). So, NeuN alone is not enough to conclude the presence of newborn neurons. Thus, along the manuscript, we specify that we analyzed cell proliferation or intermediate stages of the neurogenic process (doublecortin cells).”
Q12. Evidence is needed for data in Table 1
*ANSWER FIRST-ROUND OF REVIEW
A12. We thank this observation. In the new version of our manuscript, we included the next paragraph and tables related to the eigenvalues.
Finally, we performed the principal component analysis (PCA, Figure 6). We assembled a comprehensive set of five behavioral and four physiological parameters (Methods). The PCA revealed two main eigenvalues greater than 1 (Table 2). The PCA gave two main components explaining 81.9% variation (PC1 62.9%, PC2 19%) (Table 3; Figure 6). The PCA confirms the effect of treatment on behavioral and physiological values (Figure 6). Ki67-cells, infrapyramidal mossy fibers (CB IFPMF), time in the center of the open field (OF center time), time in the periphery of the open field (OF periphery time), and DCX- cells loaded strongly and concordantly on PC1 (62.9%, Table 3) indicating that PC1 primarily represents variation in anxiety behavior and proliferative and immature cells. PC2 (19%, Table 3) represents FST immobility time, EPM open arms time, and EPM close arms time; this may indicate a change in behavior with the TMZ-Mel treatment. The Kaiser-Meyer-Olkin (KMO) revealed the appropriate use of the factorial analysis (0.51). Also, Bartlett’s test of homogeneity of variances was significant (BKs=134.1, d.f.=8, p<2.2x10-16), suggesting a significant and relevant correlation between the variables and the factor model.
**ANSWER SECOND-ROUND OF REVIEW
From the revision of the original version, we understand that this reviewer required the information supporting the use of the principal component analysis. We hope that this answer was the expected by this reviewer.
Q13. Discussion is too long and did not focus on the study results.
*ANSWER FIRST-ROUND OF REVIEW
A13. We thank this observation because it allowed us to reorganize the discussion section.
**ANSWER SECOND-ROUND OF REVIEW
We reorganized the discussion section to focus it on the results. The discussion was separated in two sections. We could not short it because the relevant information requested by this reviewer in the abstract about the mechanisms that may explained the effects here seen should be eliminated.
Q14. What was the age and weight of mice at the begin of the study? how many mice in each group?
*ANSWER FIRST-ROUND OF REVIEW
A14. We included the missing information in the materials and methods section.
**ANSWER SECOND-ROUND OF REVIEW
The information included in the first round was (Section 4.1 animals): Balb/C female mice were ten weeks old, with 20 + 2 g used in this study.
Q15. what was the source of the drug?
*ANSWER FIRST-ROUND OF REVIEW
A15. We included the missing information in the materials and methods section.
**ANSWER SECOND-ROUND OF REVIEW
The information included in the first round was (Section 4.2 melatonin, temozolomide and experimental design):
Melatonin (Sigma, Naucalpan, Estado de México, México) was freshly prepared daily and dissolved in a minimum volume of pure ethanol plus saline solution (0.9% NaCl) with the final working concentration of 8 mg/kg body weight. The final volume of ethanol in the melatonin vehicle was less than 1% in a ratio of 1:100 (ethanol:saline solution) [20]. TMZ (Sigma, Naucalpan, Estado de México, México) was prepared in saline solution (PISA, Guadalajara, Jalisco, México) with a final 25 mg/kg concentration. The TMZ treatment followed a discontinuous cyclic administration as in preclinical studies showing depressive-like behavior and decreased hippocampal neurogenesis [8, 45].
Q16. what was the ration of ethanol & saline in melatonin vehicle.
*ANSWER FIRST-ROUND OF REVIEW
A16. We included the missing information: “Melatonin (Sigma, Naucalpan, Estado de México, México) was freshly prepared daily and dissolved in a minimum volume of pure ethanol plus saline solution (0.9% NaCl) with the final working concentration of 8 mg/kg body weight. The final volume of ethanol in the melatonin vehicle was less than 1% in a ratio of 1:100 (ethanol:saline solution).”
** ANSWER SECOND-ROUND OF REVIEW
From the revision of the original version this reviewer required the information of the ethanol & saline ratio (please see above). In the first revision of the revised version, this reviewer said: “Using alcohol as a vehicle in anxiety study is NOT correct due to its anxiolytic action.”
Regarding to the latter comment of reviewer 2, we search for papers investigating the time course of ethanol effects on anxiety and we found a study reporting that 2.4 g/kg dose caused anxiolytic effects persisting for over 2 hours after administration, effect that was not seen at longer time (Durcan MJ, Lister RG. Time course of ethanol's effects on locomotor activity, exploration and anxiety in mice. Psychopharmacology (Berl). 1988;96(1):67-72. doi: 10.1007/BF02431535. PMID: 2906444).
In our experimental design, we clearly stated that “forty-eight hours after the last administration of melatonin or vehicle, rodents underwent behavioral tests. The sequence of behavioral tests was from day 26 to day 29, mice were exposed to a rotarod (D26) and a battery of behavioral tests (open field, elevated plus maze, and Porsolt's forced swimming test, respectively.”
In this regard, it would be helpful to our present and future studies if this reviewer recommends references in which the administration of less of 1% in a ratio of 1:100 /ethanol-saline solution produces anxiolytic action when the behavior was analyzed 48 hours after the last administration.
Also, we consider important to mention that In the original and first revised version, we omitted to include the results of the rotarod. The results are mentioned in the second revised version. The results included are: “We first exposed mice to the rotarod to discard motor coordination alterations produced by treatments. We did not find differences among the groups (one-way ANOVA: F3, 29=0.10, p=0.95), suggesting that temozolomide, melatonin, or the vehicle are not affecting locomotor activity.”
Q17. In methods: write and confirm that you highlighted every possible comparison between the study groups.
*FIRST ROUND OF REVIEW
A17. In the materials and methods section (4.7), we included the phrase: “All possible comparisons among the study groups were performed.”
Q18. Were the animals trained on or Naive to the behavioral tasks? this is very important to be mentioned?
*FIRST ROUND OF REVIEW
A18. Animals were naïve to the behavioral tests. In the materials and methods section (4.3), we included the phrase: “animals were naïve to the behavioral tests.”
Q19. In page 16, how animals were euthanized? this is importent and worth mention
*FIRST ROUND OF REVIEW
A19. Mice were euthanized by decapitation two hours after the behavioral test. This information was included in the new version of our manuscript.
Q20. Mention at what data "When the normality test failed" and in figure legends mention what was the test applied to each set of data
*FIRST ROUND OF REVIEW
A20. We apologize for the inclusion of the phrase mentioned above by mistake. We have corrected the text, and in the figure legends, we mention the test applied to each data set.
**SECOND ROUND OF REVIEW
A20. We reviewed at all the statistical analysis. We incorporated the next information in the legend of figure 1: Results were analyzed with a one-way ANOVA followed by the Bonferroni post hoc test. But, when the normality test failed a Kruskal-Wallis one-way analysis of variance on ranks followed by Student-Newman-Keuls post hoc was applied (panel G, open arms time). Differences were considered statistically significant at p≤0.05. The statistical information was also included in the statistical analysis section and in the rest of figure legends.
Q21. Methods in general lacks references at many occasions.
*FIRST ROUND OF REVIEW
A21. We corrected it. Please see materials and methods section.
Q22. Use appropriate abbreviations for minutes, seconds...etc
*FIRST ROUND OF REVIEW
A22. We corrected it. Please see materials and methods section
Q23. Mention "n" in each illustration individually
*FIRST ROUND OF REVIEW
A23. We corrected it. “n” number per group was included in all figures as indicated by this reviewer.
Q24. Authors should give the source of chemicals, kits and antibodies completely and consistently (code, company, town, state and country) & version for software
*FIRST ROUND OF REVIEW
A24. We corrected it. Please see materials and methods section.
We hope that the extended answers and the new information provided respond to the observations of reviewer 2.
